# Exploration Implies Data Augmentation: Generalisation in Contextual MDPs

## Abstract

In the zero-shot policy transfer (ZSPT) setting for contextual Markov decision processes (MDP), agents train on a fixed set of contexts and must generalise to new ones. Recent work has argued and demonstrated that increased exploration can improve this generalisation, by training on more states in the training contexts. In this paper, we demonstrate that training on more states can indeed improve generalisation, but can come at a cost of reducing the accuracy of the learned value function which should *not* benefit generalisation. We hypothesise and demonstrate that using exploration to increase the agent's coverage *while also* increasing the accuracy improves generalisation even more. Inspired by this, we propose a method *Explore-Go* that implements an exploration phase at the beginning of each episode, which can be combined with existing on- and off-policy RL algorithms and significantly improves generalisation even in partially observable MDPs. We demonstrate the effectiveness of Explore-Go when combined with several popular algorithms and show an increase in generalisation performance across several environments. With this, we hope to provide practitioners with a simple modification that can improve the generalisation of their agents.

## 1 Introduction

One of the remaining challenges for developing reliable reinforcement learning (RL) agents is their ability to generalise to novel scenarios, that is, those not encountered during training. This is the main research question investigated in the zero-shot policy transfer setting (ZSPT, Kirk et al., 2023). Here the agent trains on several variations of an environment, defined by contexts, and must generalise to new ones. Recent work by Jiang et al. (2023) has demonstrated that for off-policy approaches, continuous exploration throughout the training process can improve generalisation to unseen contexts. They argue that this exploration allows the agent to learn about parts of the training environments which are not needed for the optimal policy during training, but could nonetheless improve performance when testing on unseen contexts.

This paper shows that while training on more states in the training environments can indeed improve generalisation, it can also increase the error between the learned and optimal value. This effect is demonstrated in Figure 1, where we artificially increase the number of seen states by adding random transitions (green) to a Deep Q Network replay buffer (DQN, Mnih et al., 2015, red). While the training performance remains virtually unchanged, generalisation to unseen contexts improves slightly (dashed lines). However, the right plot of Figure 1 also shows that this increases the average value error across all states in the training contexts. This reveals an interesting paradox. On the one hand, training on more transitions can reduce the number of spurious correlations between the observations and target values.[1] On the other hand, wrong targets (a common problem when training on off-policy data, Levine et al., 2020) can also introduce new spurious correlations, which should *hurt* generalisation.[2] While more exploration seems to be a net benefit in Figure 1, it begs the question whether we can further improve generalisation by maintaining the accuracy of the value function.

To address this, we introduce *reachability* to the ZSPT setting: a reachable state is any state we could encounter during training. *Unreachable generalisation* then refers to the agent's ability to generalise

---

[1] By spurious correlations we mean correlations between the observations and policy/value targets that are present in a training dataset, but not in the true underlying distribution we are trying to estimate.

[2] The correlation between input and incorrect target is not part of the true distribution, and therefore spurious.

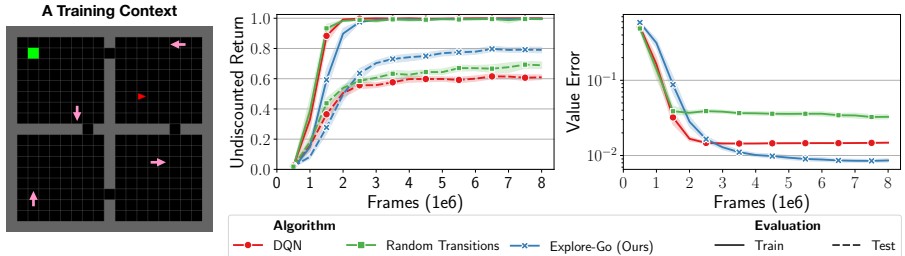

Figure 1: Left: A training context in the Four Rooms environment with examples of random transitions (pink arrows). Middle: train and test performance for DQN (red), DQN with a small fraction of random transitions added to the replay buffer (green), and DQN+Explore-Go (our method, blue). Right: the error between the estimated Q-value and the optimal value, averaged over all states in the training contexts. Mean and 95% confidence intervals over 20 seeds.

to states that are not reachable. We argue that training on more states in the training contexts results in a form of implicit data augmentation that improves the agent's unreachable generalisation performance, but that this augmentation is less effective when the target values for these additional states are not *accurate* (for example, due to bootstrapping state-action pairs that have never been seen, or that have been trained by an inaccurate target value themselves).[3]

Based on this, we introduce a method called *Explore-Go* that uses exploration at the start of each episode, in order to train on more states in the environment while also achieving a low value error, significantly increasing generalisation performance (see blue in Figure 1). Our contributions are the following:

- We demonstrate that increasing the agent's coverage of the training contexts through exploration can improve generalisation but can also result in a larger error between the learned and optimal value. Through the lens of data augmentation, we argue that training on additional states is less effective when the target values are not accurate, and we demonstrate that generalisation can be further improved by expanding coverage *while also* taking care to maintain low value error.
- We propose a method called *Explore-Go* that, to the best of our knowledge, is the first approach explicitly using exploration to improve generalization that can also be used in on-policy algorithms. Explore-Go artificially increases the number of contexts on which the agent trains by leveraging an exploration phase at the beginning of each episode. We show that it can improve generalisation performance when combined with on- and off-policy methods on several benchmarks, including both full and partial observability.[4]

## 2 BACKGROUND

A Markov decision process (MDP) $\mathcal{M}$ is defined by a 6-tuple $\mathcal{M} = \{S, A, R, T, p_0, \gamma\}$. In this definition, $S$ denotes a set of states called the state space, $A$ a set of actions called the action space, $R : S \times A \to \mathbb{R}$ the reward function, $T : S \times A \to \mathcal{P}(S)$ the transition function, where $\mathcal{P}(S)$ denotes the set of probability distributions over states $S$, $p_0 : \mathcal{P}(S)$ the starting state distribution and $\gamma \in [0, 1)$ a discount factor. The goal is to find a policy $\pi : S \to \mathcal{P}(A)$ that maps states to probability distributions over actions in such a way that maximises the expected cumulative discounted future reward $\mathbb{E}_\pi[\sum_{t=0}^\infty \gamma^t r_t]$, also called the *return*. The expectation $\mathbb{E}_\pi$ is over the Markov chain $\{s_0, a_0, r_0, s_1, a_1, r_1...\}$ induced by policy $\pi$ when acting in MDP $\mathcal{M}$ (Akshay et al., 2013). An optimal policy $\pi^*$ achieves the highest possible return. The on-policy distribution $\rho^\pi : \mathcal{P}(S)$ of the Markov chain induced by policy $\pi$ in MDP $\mathcal{M}$ defines the proportion of time spent in each state as the number of episodes in $\mathcal{M}$ goes to infinity (Sutton & Barto, 2018). Finding the optimal policy often requires to estimate the *value* or *Q-value* of a policy $\pi$, i.e. $\forall s \in S, a \in A : V^\pi(s) := \mathbb{E}\left[R(s,a) + \gamma V^\pi(s') \big|_{s' \sim T(s,a)}^{a \sim \pi(s)}\right]$ or $Q^\pi(s,a) := R(s,a) + \gamma \mathbb{E}\left[Q^\pi(s',a') \big|_{a' \sim \pi(s')}^{s' \sim T(s,a)}\right]$. Sometimes the agent does not have access to a full state, only to observations $o \sim O(s)$ that are emitted from

---

[3]We use a very broad definition of data augmentation, that includes any approach that artificially inflates the training dataset size in order to improve generalisation (Shorten & Khoshgoftaar, 2019).

[4]We provide code for our experiments at <redacted for review>.

the state $s \in S$. This is called a *partially observable MDP* (POMDP, Spaan, 2012). Here the action-observation history is Markovian and one can simply use the history instead of the state to estimate policies and values (Hausknecht & Stone, 2015).

A simple way to explore the environment is $\epsilon$-greedy exploration: $\pi_\epsilon(a|s) = (1 - \epsilon)\pi(a|s) + \epsilon\frac{1}{|A|}$, which chooses a random action with probability $\epsilon$, and follows the current policy $\pi$ otherwise. Other exploration approaches use an intrinsic reward $\eta(s, a)$ that determines what parts of the state-action space are worth exploring most (Bellemare et al., 2016; Osband et al., 2016; Ostrovski et al., 2017; Osband et al., 2018; Burda et al., 2019; Ladosz et al., 2022; Zanger et al., 2024). These intrinsic rewards can be added to the reward defined by the original MDP: $\bar{R}(s, a) = R(s, a) + \beta * \eta(s, a)$, where $\beta$ is an exploration coefficient that determines how much the agent should explore. The intrinsic reward can, for example, be based on inverse counts: $\eta(s, a) = N(s, a)^{-\frac{1}{2}}$, where $N(s, a)$ is the number of times action $a$ has been taken in state $s$ so far. This count can be *global*, meaning it keeps the count over the entire training process, or *episodic*, meaning it only counts the occurrence of a state-action pair in the current episode. When exact counting is not practical, episodic counts can be approximated with elliptical episodic bonuses (E3B, Henaff et al., 2022).

### 2.1 CONTEXTUAL MARKOV DECISION PROCESS

A contextual MDP (CMDP, Hallak et al., 2015) is a specific type of MDP where the state space $S = S' \times C$ can in principle be factored into an underlying state space $S'$ and a context space $C$, which affects rewards and transitions of the MDP. For a state $s = (s', c) \in S$, the context $c$ behaves differently than the underlying state $s'$ in that it is sampled at the start of an episode (as part of the distribution $p_0$) and remains fixed until the episode ends. The context $c$ can be thought of as the task, goal and/or environment an agent has to solve. In general, the context can influence the rewards, observations, transitions and/or the initial state distribution in $S'$. Several existing fields of study like multi-goal RL (context influences reward, Schaul et al., 2015; Andrychowicz et al., 2017) or sim-to-real transfer (context influences dynamics and/or visual observations, Tobin et al., 2017) can be framed as special instances of the CMDP framework.

The zero-shot policy transfer (ZSPT, Kirk et al., 2023) setting for CMDPs $\mathcal{M}|_C$ is defined by two disjoint sets of contexts $C^{train}, C^{test} \subset C$, $C^{train} \cap C^{test} = \varnothing$, sampled from the same distribution $\mathcal{P}(C)$ over context space $C$. The goal of the agent is to maximise performance in the testing CMDP $\mathcal{M}|_{C^{test}}$, defined by the CMDP induced by the testing contexts $C^{test}$, but the agent is only allowed to train in the training CMDP $\mathcal{M}|_{C^{train}}$. The learned policy is expected to perform *zero-shot* generalisation for the testing contexts, without any fine-tuning or adaptation period.

Recently Jiang et al. (2023) demonstrated that continuous exploration throughout training can improve generalisation to test contexts. One of their core algorithmic contributions, *temporally equalised exploration* (TEE), assigns different exploration coefficients $\beta$ to parallel workers collecting rollouts (Horgan et al., 2018), enabling continuous exploration throughout training. They also use ensembles and distributional RL in conjunction with UCB (Lattimore & Szepesvari, 2017) for better exploration. To separate the effects of *continuous* and *better* exploration, this paper uses a TEE variant where parallel workers follow *count-based intrinsic reward* with different $\beta$.

## 3 HOW EXPLORATION CAN IMPROVE GENERALISATION

The CMDP framework is very general, and the context $c \in C$ can influence several aspects of the underlying MDP, like the reward function or dynamics of the environment. To simplify analysis and facilitate cleaner definitions, this section considers a specific subset of CMDPs where the context determines only the starting state of the episode and does not influence anything beyond that. In other words, $p_0((s', c)) = p(c)p(s'|c)$, but $T((s', c), a) = T(s', a)$ and $R((s', c), a) = R(s', a)$. This might sound restrictive, but note that context-dependent transitions and rewards are still possible, just not for the same underlying state $s'$. Any CMDP where the context is observable in $s'$ is in this subset, but multiple contexts can also *share* states $s'$. Many interesting generalisation problems are captured by this setting, including environments from ViZDoom, DeepMind Control Suite and Minigrid (Wydmuch et al., 2019; Tassa et al., 2018; Chevalier-Boisvert et al., 2023).

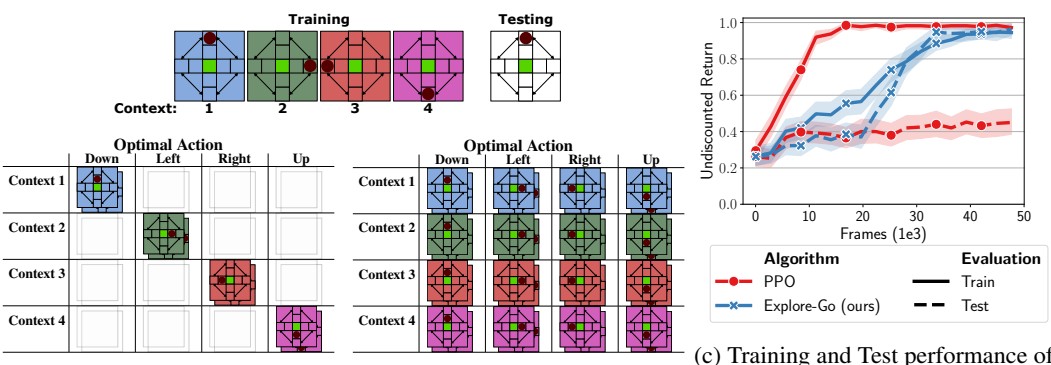

(a) States along optimal trajectory    (b) All reachable states    (c) Training and Test performance of PPO vs. PPO+Explore-Go (ours).

Figure 2: Illustrative CMDP with four training contexts (top), each with a different background colour and starting position (circle). All contexts must reach the green square in the middle. Testing is done in an unreachable context with completely different background colour (white). (a) The optimal trajectories, here sorted by their context (rows) and their optimal action (columns), show a clear spurious correlation with the background colour. (b) This correlation vanishes when training on all reachable states. (c) Performance of a baseline PPO agent and our Explore-Go agent on the CMDP. Shown are the mean and 95% confidence interval over 100 seeds.

Because we restrict ourselves to contexts that do not directly affect rewards $R$ or transitions $T$, any learned control or estimated returns learned in an underlying state $s'$ in one context transfer to the same underlying state in another context. To be clear, this setting also includes environments where contexts have completely disjoint state spaces. In a fully observable maze, for example, two contexts can start in different corners of a maze and share underlying states, but they might also start in different maze layouts, in which case they will *never* share the same underlying state. Any state $s' \in S'$ can be considered the starting state of a different context $c_{s'}$ (which is not necessarily part of the original context set $C$). As a result, we can (and will) refer to contexts and states or a set of contexts $C$ and a set of underlying starting states $S_0$ interchangeably.

## 3.1 REACHABILITY IN THE ZSPT SETTING

To argue when generalisation to different contexts is necessary, how exploration can benefit generalisation, and why accurate targets are important, we introduce the notion of *reachability of a context*. To do so, we first define the *reachability of states* in a given training CMDP $\mathcal{M}|_{S_0^{train}}$. A state $s$ is reachable, if there exists a policy whose probability of encountering that state during training is non-zero. In complement to reachable states, we define *unreachable* states as states that are not reachable. Because a context is defined by its starting state in our setting, we define (un)reachable contexts as contexts that start in a(n) (un)reachable state.

Learning from more reachable states trivially improves the performance of ZSPT to reachable contexts, as those share the exact same reachable states as the training contexts (for a more formal explanation, see Appendix B). Therefore, we only consider the setting where the agent has to generalise to unreachable contexts. This setting implies that all other states encountered in $\mathcal{M}|_{S_0^{test}}$ are also unreachable.[5] Now, to perform well during testing, the agent *has* to generalise to states it can never encounter during training. This makes it meaningfully distinct from generalisation to reachable contexts (see Appendix B for more discussion). Note that we still assume *in-distribution* generalisation, since the starting states for both train and (unreachable) test contexts are sampled from the same distribution. However, because the agent only trains on a limited set of contexts, and the test contexts can be very different from the ones seen during training, this in-distribution generalisation can be very challenging.

**Insight:** Training on more reachable states trivially improves generalisation to reachable contexts, but can it also improve generalisation to unreachable ones?

---

[5]In some non-ergodic CMDPs, it is possible to transition from an unreachable state into a reachable one, but we do not consider that in this paper.

## 3.2 GENERALISATION TO UNREACHABLE CONTEXTS

When generalising to unreachable contexts, the states encountered in $\mathcal{M}|_{S_0^{test}}$ can *never* be encountered during training. This means it is not obvious why we see exploration improving generalisation, like in Jiang et al. (2023). To investigate this, we define an example CMDP in Figure 2. This CMDP consists of a cross-shaped grid world with additional transitions that directly move the agent between adjacent end-points of the cross. The goal for the agent (circle) is to move to the centre of the cross (the green square). The four training contexts differ in the starting location of the agent and background colour. In Figure 2a the states from the *optimal* trajectories are placed in the table according to what context they are from (row) and what action is optimal (column).

Let us assume an agent does not explore extensively or stops exploring towards the end of training (due to decaying exploration rate). This agent converges to training only on the states along the optimal trajectories (assuming it solves the MDP), depicted in Figure 2a. Along these states, the background colour is perfectly correlated with the optimal action, so the agent's policy likely overfits to this spurious correlation. As a result, this policy is unlikely to generalise to a new *unreachable* context with an unseen background colour (i.e., the testing context at the top of Figure 2). We show this empirically in Figure 2c, where a PPO agent (red) learns all training contexts perfectly, but does not generalise to a new background colour (dashed line, see Appendix C.1 for more).

Suppose now, we have a policy that has learned over the entire reachable state space (see Figure 2b), not just the optimal trajectories. This agent is more likely to ignore the background colour, as it no longer correlates with the optimal action. We see this ability to uncover the true relationships and generalise to new colours when training on all reachable contexts (blue in Figure 2c, our approach further introduced in Section 4). We can view the inclusion of additional reachable states as a form of data augmentation. For example, the additional states from contexts 2, 3 and 4 in the first column in Figure 2b, can be viewed as simple visual transformations of the state from Context 1 that do not affect the underlying meaning.

This idea is not limited to this example; the observable states of many environments will contain patterns that spuriously correlate with the desired target outputs *only* on the optimal trajectories. Data augmentation has proven to be a powerful tool for addressing such spurious correlations and improving generalisation performance across a wide range of settings and applications (Shorten & Khoshgftaar, 2019; Feng et al., 2021; Zhang et al., 2021a). Its benefits have been attributed to reducing the probability of overfitting and/or regularising training, and extend far beyond the invariance to background colour as illustrated in the example of Figure 2. Generalisation improvements have been observed even when the augmentations are not consistent with the data distribution (such as random croppings, mixing of features, or even random convolutions) (Wu et al., 2020; Raileanu et al., 2021; Lin et al., 2022; Geiping et al., 2023), or when they are only applied to specific classes (Miao et al., 2023; Shen et al., 2022). As such, we can view the use of exploration to artificially increase the number of states on which the agent trains as a form of implicit data augmentation, and expect the induced generalisation benefits argued above to extend to more complex CMDPs.

Note that while the explanation here is for fully observable CMDPs, we can extend it to partially observable environments. The agent then trains on histories rather than states, but a similar argument exists: increasing the coverage of histories should improve generalization to new ones. We observe this when evaluating our method in a partially observable ViZDoom environment in Section 5.3.

> **Insight:** Generalisation to unreachable contexts *can* be improved by performing data augmentation in the form of training on more reachable contexts/states.

## 3.3 THE ISSUE WITH EXPLORATION-INDUCED DATA AUGMENTATION

Extended exploration (as in Jiang et al., 2023) chooses trajectories that visit more states, but those often provide poor target estimates. For example, estimating the Q-value $Q^\pi(s, a)$ of the policy $\pi$ requires bootstrapping with the Q-value $Q^\pi(s', a')$ of the next state $s' \in S$ and the action(s) $a' \in A$ that $\pi$ would choose in $s'$. To get an accurate estimate of $Q^\pi(s', a')$, we need to observe the consequences of choosing $a'$ in $s'$, but exploration often picks actions the policy would not. Therefore, bootstrapping from state-actions that are not trained on makes the targets of many exploratory transitions inaccurate. Even if the policy's actions $a'$ are explored in $s'$, their Q-value is bootstrapped with *their* next state and so on. For the states infrequently visited by exploration, the learned $Q^\pi(s, a)$ are thus unlikely to

be accurate. One way to ensure this value estimate is accurate would be to follow the trajectory of policy $\pi$ and learn from the gathered transitions (see Section 4).

Nonetheless, even with poor targets there can still be some benefit from training on the additional states that extensive exploration provides. For example, in Figure 2b, even if the target would be wrong for some of the states (the policy learns an action other than the optimal one), the spurious correlation with background colour would be broken, which might improve generalisation. However, to fit the wrong targets, the agent likely has to overfit to some newly introduced spurious correlations involving a combination of background colour and agent position, which might hurt generalisation.

> **Insight:** Exploration can introduce inaccurate targets which could hurt unreachable generalisation.

In conclusion, training on more data can reduce the spurious correlations in the dataset, but training with incorrect labels can introduce new spurious correlations. Which of these two will influence generalisation performance more depends on the environment's input features. In the Four Rooms example in Figure 1, the benefits of training on additional random transitions seem to outweigh the disadvantages of incorrect labels slightly. However, to improve this trade-off, we propose to leverage exploration in such a way that the agent trains on as many reachable states as possible *while also* maintaining accurate targets for these states.

## 4 EXPLORE-GO: TRAINING ON MORE REACHABLE CONTEXTS WITH ACCURATE TARGETS

To improve generalization by both training on additional states *and* providing accurate targets for these states, we propose them as the starting states of additional training contexts. By doing so, we can rely on the RL algorithm to find an optimal policy and converge to on-policy, optimal trajectories from these states, resulting in accurate targets for our value and/or policy function. We propose a method *Explore-Go*[6] which effectively trains in more reachable contexts by artificially increasing the diversity of the starting state distribution. It achieves this by introducing an exploration period at the start of each training episode. We effectively leverage the versatile literature on exploration approaches to implicitly generate new reachable contexts for the agent to train on. At no point does Explore-Go require explicit knowledge of which states/contexts are reachable.

Our method modifies a core part of RL algorithms: the collection of rollouts. At the start of every episode, Explore-Go first enters a phase in which it explores the environment by following a *pure exploration* policy (Bubeck et al., 2009). Pure exploration refers to an objective that ignores the rewards $r_t$ the agent encounters and instead focuses on exploring new parts of the state space. We aim to solve the CMDP for as many reachable contexts as possible. For a more even distribution, we interrupt the pure exploration phase after $k$ steps, where $k$ is drawn uniformly between 0 and $K$. Whatever state the pure exploration phase ends in will be treated by the agent as the starting state of that episode. This includes doing any exploration that the agent might normally perform, and resetting the history in partially observable environments.

We only train our agent on the experience collected after the pure exploration phase. The pure exploration experience will likely not have accurate targets and can therefore hurt generalisation (see Appendix E.2). An example of a generic rollout collection protocol modified with Explore-Go can be found in Algorithm 1 in Appendix D. Note that this does reduce the sample efficiency of Explore-Go, but should nonetheless result in better generalisation performance. The pure exploration experience *can* be used to train a separate pure exploration agent in parallel to the main agent (as demonstrated with PPO in Section 5.1). See Algorithm 2 in Appendix D for example pseudo-code.

Even though Explore-Go changes the distribution of the training data, it can be combined with both off-policy *and* on-policy reinforcement learning methods. On-policy approaches typically require (primarily) transitions sampled by the current policy $\pi_\theta$ for training. This means they would not work with arbitrary changes to the training data distribution. However, Explore-Go effectively only changes the distribution of the starting states $S_0^{train}$. So, we can think of Explore-Go as generating on-policy data for a modified MDP that differs only in its starting state distribution.

---

[6]The name Explore-Go is a variation of the popular exploration approach Go-Explore (Ecoffet et al., 2021). In Go-Explore the agent teleports at the start of each episode to a novel state and then continues exploration. In our approach, the agent first explores and then goes and solves the original environment.

## 5 EXPERIMENTS

We perform an empirical evaluation of the unreachable generalisation performance of Explore-Go on environments from three benchmarks: Four Rooms from Minigrid (Chevalier-Boisvert et al., 2023), My Way Home from ViZDoom (Wydmuch et al., 2019) and Finger Turn and Reacher from the DeepMind Control Suite (DMC, Tassa et al., 2018). For each of the experiments in this section, we trained an agent on a small set of contexts and evaluated the agent's generalisation to unreachable ones (see Section 3.1 for a definition of reachability). Note that Explore-Go does not perform pure exploration at the start of testing episodes, but only during training.

Due to its discrete nature and smaller size, we use the Four Rooms environment to demonstrate the versatility of Explore-Go. Since we can enumerate all possible states and contexts and formulate optimal policies and values, we also use it to analyse our method and compare it to other baselines. Across all experiments, we evaluate Explore-Go combined with several on-policy, off-policy, value-based and/or policy-based RL algorithms: PPO (on-policy, policy-based), DQN (off-policy, value-based), soft actor-critic (SAC, off-policy, policy-based, Haarnoja et al., 2018) and SAC with augmented visual data (RAD, Laskin et al., 2020).

### 5.1 EXPLORE-GO WITH OFF- AND ON-POLICY ALGORITHMS

We evaluate in the Four Rooms environment from Minigrid, modified to be fully observable and have a reduced action space (see Figure 6 and Appendix C.2). This environment is a grid-world containing four rooms with single-width doorways connecting all of the rooms. The agent starts in one of the rooms and must move to the goal location, which may be in a different room. Contexts differ from each other in the starting location and orientation of the agent, the goal location, and the position of the doorways connecting the four rooms. In our experiments, the agent trained on 200 different training contexts and was evaluated on 200 unreachable contexts. A Four Room's context is reachable if and only if both the positions of the doorways and the goal location are the same as at least one training context. We used deep exploration for the baselines, and both the main agent and the pure exploration phase of Explore-Go. The intrinsic reward is based on a combination of episodic and global state-action counts similar to prior work on exploration in CMDPs (Henaff et al., 2023; Flet-Berliac et al., 2021; Samvelyan et al., 2021; Zhang et al., 2021c). For PPO+Explore-Go specifically, we trained a separate pure exploration agent in parallel, as discussed in Section 4. We refer to Appendix C.2 for more experimental details on the Four Rooms experiments.

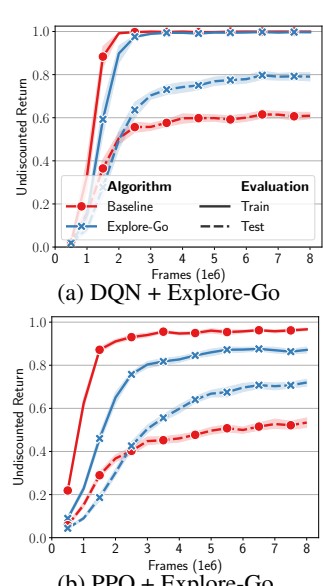

(a) DQN + Explore-Go

(b) PPO + Explore-Go

Figure 3: Training and unreachable test performance in Four Rooms. Mean and 95% confidence intervals for 20 seeds.

In Figure 3 we see that Explore-Go improves the testing performance on unreachable contexts when combined with DQN and PPO, despite worse sample efficiency and PPO training performance. More experiments supporting this conclusion with SAC and RAD in the continuous control DMC environments can be found in Appendix E.1. In Appendix E.3 we also show that test performance is not very sensitive to the specific value of the hyperparameter $K$ introduced by Explore-Go.

**Conclusion:** Explore-Go can significantly improve generalisation to unreachable contexts when combined with on- or off-policy algorithms.

### 5.2 THE EFFECT OF DIFFERENT EXPLORATION APPROACHES ON GENERALISATION

Explore-Go implicitly creates additional contexts on which the agent trains. We argue that this trades-off data augmentation with more reachable states versus more accurate targets, leading to improved generalisation. To disentangle these two effects, we compare Explore-Go with continuous exploration throughout training (TEE, Jiang et al., 2023, see Section 2.1). To the best of our knowledge, this prior approach is the state-of-the-art for methods that specifically use exploration to improve generalisation. TEE only works with off-policy methods, so we compare it with the DQN+Explore-Go agent from the previous section. More information on TEE is provided in Appendix C.2.

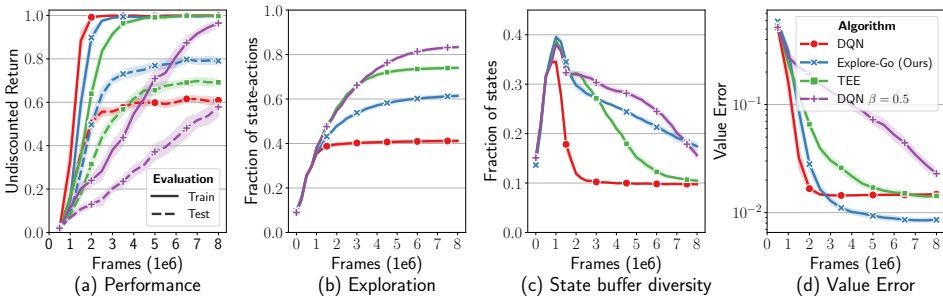

Figure 4: Comparing DQN, DQN+Explore-Go, DQN+TEE and DQN with increased exploration ($\beta = 0.5$) in Four Rooms: (a) train and test performance, (b) fraction of reachable state-action space explored, (c) fraction of reachable state space currently in the buffer and (d) true value error averaged of the entire reachable state space. Shown are the mean and 95% confidence intervals over 20 seeds.

In Figure 4 we show that despite exploring a larger fraction of the state-action space (Figure 4b), TEE generalises *worse* than Explore-Go (Figure 4a). Additionally, TEE has a significantly worse value error than Explore-Go and converges relatively slowly to a similar error as the baseline (Figure 4d). We observe a similar trend if the exploration coefficient $\beta$ for the baseline DQN is increased (from $\beta = 0.01$ to $\beta = 0.5$). This *does* explore more of the state-action space (Figure 4b), increases the diversity of the replay buffer (Figure 4c), but *does not* result in a more accurate value function (Figure 4d), and yields worse sample efficiency and generalisation than Explore-Go (Figure 4a). This suggests a trade-off between exploration and value error, and that simply increasing exploration does not necessarily lead to better generalisation. See Appendix C.2 for more on these metrics.

> **Conclusion:** It appears generalisation is not only about *how much* the agent explores or *how diverse* the training data is, but also *how accurate* the values in the explored states are.

### 5.3 EXPLORE-GO ON PARTIALLY OBSERVABLE ENVIRONMENTS

We have demonstrated the ability of Explore-Go to improve generalisation to new contexts in fully observable environments. Here we consider the more challenging partially observable setting with first-person image observations from ViZDoom (see top of Figure 5, Wydmuch et al., 2019). We evaluate the My Way Home scenario, where the agent must navigate through a labyrinth-like surrounding to reach a green vest. We turn this environment into a CMDP by having the context determine the goal location and the initial position and orientation of the agent. Our baseline agent uses the Asynchronous PPO (APPO) algorithm as implemented by Sample Factory (Petrenko et al., 2020). APPO+Explore-Go uses an E3B (Henaff et al., 2022) pure exploration agent trained in parallel (as described in Section 4). In Figure 5 we see APPO and APPO+Explore-Go achieve similar training performance, but Explore-Go significantly improves the generalisation to unseen, unreachable contexts. See Appendix C.3 for more details.

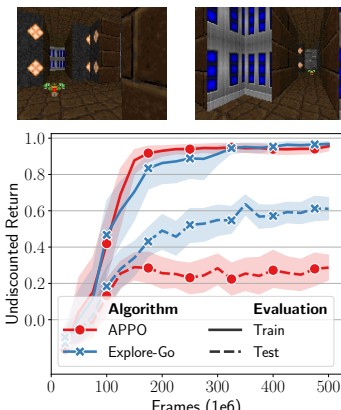

Figure 5: The partially observable ViZDoom "My Way Home" CMDP (top). Mean and 95% confidence intervals for 20 seeds.

## 6 RELATED WORK

The contextual MDP framework encompasses many fields that study zero-shot generalisation. Some approaches try to improve generalisation by increasing the variability of the training contexts through domain randomisation (Tobin et al., 2017; Sadeghi & Levine, 2017) or data augmentation (Raileanu et al., 2021; Lee et al., 2020). Others explicitly bridge the gap between the training and testing contexts through inductive biases (Kansky et al., 2017; Wang et al., 2021) or regularisation (Cobbe et al., 2019; Tishby & Zaslavsky, 2015). We mention only a selection of approaches here, a more comprehensive overview and discussion can be found in Appendix A.1. All these approaches use techniques that are not necessarily specific to RL (representation learning, regularisation, etc.). In this work, we instead explore how exploration in RL can be used to improve generalisation.

Next, we discuss related work on exploration in CMDPs. Zisselman et al. (2023) leverage exploration at test time to move the agent towards states where it can confidently solve the context, thereby increasing test time performance. Our work differs in that we leverage exploration during training in order to increase the number of states from which the agent can confidently solve the test contexts. Similar to our work, Jiang et al. (2023) use exploration during training to improve generalisation. However, their approach only works for off-policy algorithms, whereas ours can be applied to both off-policy and on-policy methods. For more discussion on the differences between their work and ours, see Appendix A.2. Zhu et al. (2020) learn a reset controller that increases the diversity of the starting states. However, they only consider reachable generalisation, which is meaningfully distinct from the unreachable generalisation setting considered in this paper (see Appendix B). Suau et al. (2024) introduce the notion of policy confounding in out-of-trajectory generalisation. The issue of policy confounding is complementary to our intuition for unreachable generalisation. However, they do not propose a new, scalable approach to tackle the issue. More discussion on extended related work is given in supplemental material A.1.

# 7 DISCUSSION AND LIMITATIONS

In this paper, we extend prior work by demonstrating that training on more parts of the training environment can indeed improve generalisation (Jiang et al., 2023), but can come at a cost of increased value error. To understand why this can hurt generalisation, we define the notion of *reachability* of states and contexts. We argue that exploration can improve generalisation to unreachable contexts only indirectly from the data augmentation that comes from training on more reachable contexts. With this perspective, we make the conjecture that this data augmentation would improve generalisation the most if it was done with accurate value targets.

To validate this, we define *Explore-Go*, that trains on more states from the training contexts, whilst decreasing value error. It does so by beginning each episode with a pure exploration phase, effectively increasing the diversity of the starting states. Although an increased diversity in initial states has been linked to improved generalisation before (Zhu et al., 2020; Tobin et al., 2017), this paper contextualises this in the ZSPT setting. We argue for and show that generalisation to *unreachable* states can be improved, even if the increase in diversity only comes from adding *reachable* starting states, which are states from contexts the agent was already training on. Furthermore, we connect this approach to previous work improving generalisation through exploration, by arguing that *reachable states*, however sampled, should be treated properly as starting states in *reachable contexts*.

We demonstrate empirically that in comparison to a baseline algorithm, Explore-Go trains on more diverse states, improves value accuracy, and increases unreachable generalisation performance. This improves significantly over the previous exploration-based state-of-the-art generalisation method TEE, which explores more, but has significantly worse value accuracy, and cannot be used with on-policy approaches. In contrast, Explore-Go can be applied easily to both on-policy and off-policy methods. We demonstrate that it increases generalisation when combined with DQN, PPO, APPO, SAC and RAD, that it scales up to a more complex scenario from the partially-observable ViZDoom environments, and to state- and image-based tasks from the DeepMind Control Suite.

Although we demonstrate significant generalisation benefits across various algorithms (on- and off-policy), settings (fully and partially observable, discrete and continuous), and input modalities (images and proprioceptive states), most of our experiments could be classified as having *state-reaching* objectives. These objectives reflect many challenging and important problems encountered in the real world (Colas et al., 2020; Liu et al., 2022). Since data augmentation has shown success in many different domains and settings, we believe our approach might hold on other types of problems as well, and we see this as a promising direction of future research. Additionally, although we show generalisation benefit using both simple random exploration and more sophisticated deep exploration methods, we do not extensively analyse which pure exploration approaches would improve generalisation the most. Therefore, an interesting direction for future work is to determine ideal start state distributions for generalisation and how to generate these states through exploration.

In conclusion, we present a novel approach Explore-Go that effectively leverages exploration to increase the starting state distribution of the agent, balancing increased training diversity with accurate learning targets, resulting in improved generalisation. With this, we hope to provide practitioners with a simple modification that can improve the generalisation of their agents significantly.

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

## A RELATED WORK

### A.1 EXTENDED RELATED WORK

#### A.1.1 GENERALISATION IN CMDPS

The contextual MDP framework is a very general framework that encompasses many fields in RL that study zero-shot generalisation. One such setting is the *sim-to-real* setting often encountered in robotics (Kirk et al., 2023). In sim-to-real, the goal is to train in simulation and generalise to the real world. Since the real world can rarely be expressed by a single context, the learned policy needs to generalise out-of-distribution to achieve this. So, although sim-to-real can be described as a CMDP, it differs from the ZSPT setting assumed in our paper, which considers in-distribution generalisation. An approach used in the sim-to-real setting is domain randomisation (Tobin et al., 2017; Sadeghi & Levine, 2017; Peng et al., 2018), where the context distribution during training is significantly increased in order to train a policy that can generalise over as many different scenarios as possible. This differs from the ZSPT setting in our work in that we assume access to only a limited set of training contexts. Furthermore, domain randomisation *explicitly* generates additional (often *unreachable*) training contexts. In contrast, our work could be viewed as *implicitly* generating additional *reachable* contexts through exploration. Another approach that increases the context distribution is data augmentation (Raileanu et al., 2021; Lee et al., 2020; Zhou et al., 2021). These approaches work by applying a set of given transformations to the states with the prior knowledge that these transformations leave the output (policy or value function) invariant. In this paper, we argue that our approach implicitly induces a form of invariant data augmentation on the states. However, this differs from the other work cited here in that we don't explicitly apply transformations to our states, nor do we require prior knowledge on which transformations leave the policy invariant.

So far we have mentioned some approaches that increase the number and variability of the training contexts. Other approaches instead try to explicitly bridge the gap between the training and testing contexts. For example, some use inductive biases to encourage learning generalisable functions (Zambaldi et al., 2018; 2019; Kansky et al., 2017; Wang et al., 2021; Tang et al., 2020; Tang & Ha, 2021). Others use regularisation techniques from supervised learning to boost generalisation performance (Cobbe et al., 2019; Tishby & Zaslavsky, 2015; Igl et al., 2019; Lu et al., 2020; Eysenbach et al., 2021). We mention only a selection of approaches here, for a more comprehensive overview we refer to the survey by Kirk et al. (2023).

All the approaches above use techniques that are not necessarily specific to RL (representation learning, regularisation, etc.). In this work, we instead explore how exploration in RL can be used to improve generalisation.

#### A.1.2 EXPLORATION IN CMDPS

There have been numerous methods of exploration designed specifically for or that have shown promising performance on CMDPs. Some approaches train additional adversarial agents to help with exploration (Flet-Berliac et al., 2021; Campero et al., 2021; Fickinger et al., 2021). Others try to exploit actions that significantly impact the environment (Seurin et al., 2021; Parisi et al., 2021) or that cause a significant change in some metric (Raileanu & Rocktäschel, 2020; Zhang et al., 2021c;b; Ramesh et al., 2022). More recently, some approaches have been developed that try to generalise episodic state visitation counts to continuous spaces (Jo et al., 2022; Henaff et al., 2022) and several studies have shown the importance of this for exploration in CMDPs (Wang et al., 2023; Henaff et al., 2023). All these methods focus on trading off exploration and exploitation to achieve maximal performance in the training contexts as fast and efficiently as possible. However, in this paper, we examine the exploration-exploitation trade-off to maximise generalisation performance in testing contexts.

In Zisselman et al. (2023), the authors leverage exploration at test time to move the agent towards states where it can confidently solve the context, thereby increasing test time performance. Our work differs in that we leverage exploration during training time to increase the number of states from which the agent can confidently solve the test contexts. In Jesson et al. (2024), they use, among other things, exploration based on Thompson sampling and demonstrate improved generalisation on the Procgen benchmark. However, our work focusses on *how* exploration should be used, rather than *which* specific exploration technique works best. Closest to our work is Jiang et al. (2023), Zhu et al.

(2020) and Suau et al. (2024). Jiang et al. (2023) use extensive exploration throughout training to improve generalisation. However, their approach only works for off-policy algorithms, whereas ours can be applied to both off-policy and on-policy methods. For more discussion on the differences between their work and ours, we refer to Appendix A.2 In Zhu et al. (2020), the authors learn a reset controller that increases the diversity of the agent's start states. However, they only argue (and empirically show) that this benefits reachable generalisation, which (as argued in Appendix B) is meaningfully distinct from the unreachable generalisation setting considered in this paper. Suau et al. (2024) introduces the notion of policy confounding in out-of-trajectory generalisation. The issue of policy confounding is complementary to our intuition for unreachable generalisation. However, it is unclear how out-of-trajectory generalisation equates to reachable or unreachable generalisation. Moreover, they do not propose a new, scalable approach to solve the issue.

## A.2 DISCUSSION ON RELATED WORK

Jiang et al. (2023) argue that generalisation in RL extends beyond representation learning, by using an example in a tabular grid-world environment. In their environment the agent always starts in the top left corner of the grid during training, and the goal is always in the top right corner. During testing the agent starts in a different position in the grid-world (in their example, the lower left corner). They then argue that more exploration can improve generalisation to these contexts. According to our definitions, this is an example of generalisation to a reachable context.

They extend their intuition to non-tabular CMDPs by arguing that in certain cases two states that are unreachable from each other, can nonetheless map to similar representations inside a neural network. As a result, even though a state in the input space is unreachable, it can be mapped to something reachable in the latent representational space and therefore the reachable generalisation arguments apply. For this reason, the generalisation benefits from more exploration can go beyond representation learning.

Our paper introduces the distinction between reachable and unreachable generalisation, allowing us to focus on the latter (more complicated) setting. We argue, through the lens of implicit data augmentation, training on more of these states could encourage the agent to learn representations that generalise well to unreachable states. As such, we believe the generalisation benefits for unreachable contexts *can* in part be attributed to a form of representation learning as well. Furthermore, by focussing on the unreachable generalisation setting, we identify the issue of inaccurate learning targets, and propose our method Explore-Go to mitigate this.

## B GENERALISATION TO REACHABLE CONTEXTS

We introduce the notion of *reachability of a context*. To do so, we first define the *reachability of states* in a CMDP $\mathcal{M}|_{S_0^{train}}$. The set of reachable states $S_r(\mathcal{M}|_{S_0^{train}})$, abbreviated with $S_r$ from now on, consists of all states $s_r$ for which there exists a sequence of actions that give a non-zero probability of ending up in $s \in S_r$ when performed from at least one starting state in $\mathcal{M}|_{S_0^{train}}$.[7] Put differently, a state $s$ is reachable if there exists a policy whose probability of encountering that state during training is non-zero. In complement to reachable states, we define *unreachable* states $s \in S_u$, $\quad S_u = S \setminus S_r$ as states that are not reachable.

We also define two instances of the ZSPT problem in the subset of CMDPs where the context is fully determined by the starting state (as introduced in Section 3).

**Definition 1** (Reachable/Unreachable generalisation). *Reachable/Unreachable generalisation refers to an instance of the ZSPT problem where the start states of the testing environments $S_0^{test}$ are/are-not part of the set of reachable states during training, i.e. $S_0^{test} \subseteq S_r$ or $S_0^{test} \cap S_r = \emptyset$.*

---

[7]This definition only works for discrete states. For continuous states, we can similarly define a reachable state $s \in S_r$ as a state for which there exists a sequence of actions that gets arbitrarily close to $s$ when performed in $\mathcal{M}|_{S_0^{train}}$.

This definition has some interesting implications: due to how reachability is defined, in the reachable generalisation setting all states encountered in the testing CMDP $\mathcal{M}|_{S_0^{test}}$ are also reachable[8].

In the single-task RL setting, the goal is to maximise performance in the MDP $\mathcal{M}$ in which the agent trains. Acting optimally in all the states encountered by the optimal policy in $\mathcal{M}$ guarantees maximal return in $\mathcal{M}$. Therefore, exploration only has to facilitate learning the optimal policy in the states present in the on-policy distribution $\rho^{\pi^*}$ of $\mathcal{M}$. In fact, once the optimal policy has been found, learning to be optimal anywhere else in $\mathcal{M}$ would be a wasted effort that potentially allocates approximation capacity to unimportant areas of the state space.

Recent work has shown that this logic does not transfer to the ZSPT setting (Jiang et al., 2023). In this setting, the goal is not to maximise performance in the training CMDP $\mathcal{M}|_{S_0^{train}}$, but rather to maximise performance in the testing CMDP $\mathcal{M}|_{S_0^{test}}$. Ideally, the learned policy will be optimal over the on-policy distribution $\rho^{\pi^*}$ in this testing CMDP. In general, this testing distribution is unknown. However, in the reachable generalisation setting, the starting states during testing are (by definition) part of the reachable state space $S_r$. So, an agent that learns to act optimally in as many of the reachable states as possible can improve its performance during testing. In fact, if a policy were optimal in all reachable states, it would be guaranteed to 'generalise' to any reachable context. One could argue generalisation is not the best term to use here, since even a policy that completely overfits to the reachable state space $S_r$, for example, a tabular setting, would exhibit perfect 'generalisation'.

**Corollary 1.** *An optimal policy $\pi$ that achieves maximal return from any state in the reachable state space $S_r(\mathcal{M}|_{S_0^{train}})$, will have optimal performance in the reachable generalisation setting.*

Recall that performance in a ZSPT problem is defined as the performance in the testing MDP $\mathcal{M}|_{S_0^{test}}$, which in the case of reachable generalisation, has an underlying state space that consists only of reachable states. It follows naturally that a policy that is optimal on the entire reachable state space $S_r(\mathcal{M}|_{S_0^{train}})$ also has to be optimal in $\mathcal{M}|_{S_0^{test}}$.

Note that the conclusions above do not hold for the unreachable generalisation setting, which *does* require explicit generalisation to states it can never encounter during training. As such, these two settings are meaningfully distinct.

## C Experimental details

### C.1 Illustrative CMDP

Training is done on the four contexts in Figure 2 and unreachable generalisation is evaluated on new contexts with a completely different background colour. For pure exploration, we sample uniformly random actions at each timestep ($\epsilon$-greedy with $\epsilon = 1$). We compare Explore-Go to a baseline using regular PPO. In Figure 2c we can see that the PPO baseline achieves approximately optimal training performance but is not consistently able to generalise to the unreachable contexts with a different background colour. PPO trains mostly on on-policy data, so when the policy converges to the optimal policy on the training contexts it trains almost exclusively on the on-policy states in Figure 2a. As we hypothesise, this likely causes the agent to overfit to the background colour, which will hurt its generalisation capabilities to unreachable states with an unseen background colour. On the other hand, Explore-Go maintains state diversity by performing pure exploration steps at the start of every episode. As such, the state distribution on which it trains resembles the distribution from Figure 2b. As we can see in Figure 2c, Explore-Go learns slower, but in the end achieves similar training performance to PPO and performs significantly better in the unreachable test contexts. We speculate this is due to the increased diversity of the state contexts on which it trains.

#### Environment details

The training contexts for the illustrative CMDP are the ones depicted in Figure 2. The unreachable testing contexts consist of 4 contexts with the same starting positions as found in the training contexts (the end-point of the arms) but with a white background colour. The states the agent observes are

---

[8]Note that the reverse does not have to be true: not all reachable states can necessarily be encountered in $\mathcal{M}|_{S_0^{test}}$.

structured as RGB images with shape $(3, 5, 5)$. The entire $5 \times 5$ grid is encoded with the background colour of the particular context, except for the goal position (at $(2, 2)$) which is dark green ($(0,0.5,0)$ in RGB) and the agent (wherever it is located at that time) which is dark red ($(0.5,0,0)$ in RGB). The specific background colours are the following:

- **Training context 1:** (0,0,1)
- **Training context 2:** (0,1,0)
- **Training context 3:** (1,0,0)
- **Training context 4:** (1,0,1)
- **Testing contexts:** (1,1,1)

Moving into a wall of the cross will leave the agent position unchanged, except for the additional transitions between the cross endpoints (e.g., moving right at the end-point of the northern arm of the cross will move you to the eastern arm). Moving into the goal position (middle of the cross) will terminate the episode and give a reward of 1. All other transitions give a reward of 0. The agent is timed out after 20 steps.

IMPLEMENTATION DETAILS

For PPO we used the implementation by Moon et al. (2022) which we adapted for PPO + Explore-Go. The hyperparameters for both PPO and PPO + Explore-Go can be found in Table 1. The only additional hyperparameter that Explore-Go uses is the maximal number of pure exploration steps $K$, which we choose to be $K = 8$. Both algorithms use network architectures that flatten the $(3, 5, 5)$ observation and feed it through a fully connected network with a ReLU activation function. The hidden dimensions for both the actor and critic are $[128, 64, 32]$ followed by an output layer of size $[1]$ for the critic and size $[|A|]$ for the actor. The output of the actor is used as logits in a categorical distribution over the actions.

Table 1: Hyper-parameters used for the illustrative CMDP experiment

| **Illustrative** | |
| --- | --- |
| **Hyper-parameter** | **Value** |
| Total timesteps | 50 000 |
| Vectorised environments | 4 |
| **PPO** | |
| timesteps per rollout | 10 |
| epochs per rollout | 3 |
| minibatches per epoch | 8 |
| Discount factor $\gamma$ | 0.9 |
| GAE smoothing parameter ($\lambda$) | 0.95 |
| Entropy bonus | 0.01 |
| PPO clip range ($\epsilon$) | 0.2 |
| Reward normalisation? | No |
| Max. gradient norm | .5 |
| Shared actor and critic networks | No |
| **Adam** | |
| Learning rate | $1 \times 10^{-4}$ |
| Epsilon | $1 \times 10^{-5}$ |

## C.2 FOUR ROOMS

In our Four Rooms experiments, we train on 200 training contexts and test on either a reachable or unreachable context set of size 200. The 200 training contexts differ in the agent location, agent direction, goal location and the location of the doorways (see Figure 6 for some example contexts in Four Rooms).

In this environment, reachability is regulated through variations in the goal location and location of the doorways. If two states share their doorways and goal location, then they are both reachable

from one another. Conversely, if two states differ in either the doorways or goal location, they are unreachable. The reachable context set is constructed by taking every training context and changing only the agent location and agent direction (keeping the location of the doorways and goal location the same). For the unreachable context set, we take 200 different configurations of the doorways that all differ from the ones in the training context. For each of those 200 different doorway configurations, we generate a new goal location, agent location and agent direction.

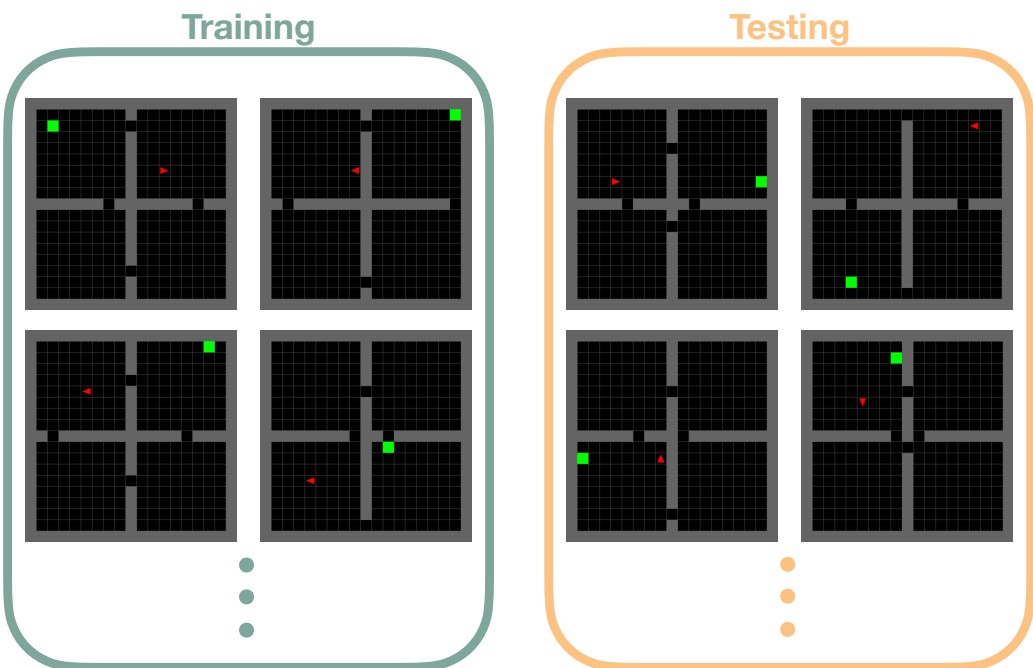

Figure 6: Some example contexts in the Four Rooms environment for unreachable generalisation.

ENVIRONMENT DETAILS

The Four Rooms grid world used in our experiments differs in certain ways from the default Minigrid configuration (Chevalier-Boisvert et al., 2023). For one, the action space is reduced from the default seven actions (turn left, turn right, move forward, pick up an object, drop an object, toggle/activate an object, end episode) to just the first three actions (turn left, turn right, move forward). Also, the reward function is changed slightly to reward 1 for successfully reaching the goal and 0 otherwise (as opposed to the $1 - 0.9 * \left(\frac{\text{step count}}{\text{max steps}}\right)$ given upon success by the default Minigrid environment).

The implementation of Four Rooms is also customised to allow for more control over the factors of variation (topology, agent location, agent direction, goal location) for generating contexts. This acts functionally the same as the ReseedWrapper from Minigrid except that it allows for more control and therefore easier design and construction of the training and testing sets. The code for our Four Room implementation can be found at <redacted for review>.

EXPLORE-GO WITH DQN AND PPO

For the DQN and PPO experiments, we take the implementations from the Stable-Baselines3 (Raffin et al., 2021) repository and add Explore-Go to them (see code at <redacted for review>). For all experiments, the network architecture consists of a small ResNet (He et al., 2016) feature extractor from IMPALA (Espeholt et al., 2018) with only a single ResNet sequence (consisting of a single convolutional layer followed by a max pool operation and two residual blocks consisting of two convolutional layers with a residual connection each). A list of encoder hyperparameters can be found in Table 2. After the encoder are two fully connected layers of sizes [512,256].

For deep exploration, we use an intrinsic reward $\eta(s, a)$ that is a multiplication of a global reward and episodic reward:

$$\eta(s, a) = \eta_{\text{global}}(s, a) * \eta_{\text{local}}(s, a)$$

$$\eta_{\text{global}}(s, a) = \frac{1}{\sqrt{N_g(s, a)}}$$

$$\eta_{\text{local}}(s, a) = \mathbb{I}[N_e(s, a) = 1]$$

where $N_g(s, a)$ is a global state-action count and $N_e(s, a)$ an episodic one. For the baseline agent of PPO, the intrinsic reward $\eta(s, a)$ is simply added to the extrinsic reward: $\bar{r}(s, a) = r(s, a) + \beta * \eta(s, a)$. For PPO+Expore-Go a pure exploration agent is trained in parallel on the experience collected during the pure exploration phase using only intrinsic rewards: $r_{\text{pure}}(s, a) = \beta_{\text{pure}} * \eta(s, a)$. For the DQN baseline, we train two estimates of the Q values: Q(s,a) for the extrinsic reward and U(s,a) for the intrinsic reward. During rollouts the DQN policy is then given by: $\pi(a|s) = \text{argmax}_a Q(s, a) + \beta * U(s, a)$. For the pure exploration phase, this means we can simply use $\pi_{pure}(a|s) = \text{argmax}_a U(s, a)$ as a pure exploration policy without any additional training required.

We perform a hyperparameter search for the DQN and PPO baselines over the following values for DQN:

- **Soft update coefficient Q**: $\{0.05, 0.01, 0.005\}$
- **Soft update coefficient U**: $\{0.05, 0.01, 0.005\}$
- **Learning rate Q**: $\{1 * 10^{-4}, 5 * 10^{-4}, 1 * 10^{-3}\}$
- **Learning rate U**: $\{1 * 10^{-4}, 5 * 10^{-4}, 1 * 10^{-3}\}$
- **Exploration coefficient** $\beta$: $\{0.5, 0.1, 0.05, 0.01\}$

and for PPO:

- **Number of training epochs per rollout**: $\{1, 5, 10\}$
- **Entropy coefficient**: $\{0.0, 0.01, 0.001\}$
- **Clip range**: $\{0.1, 0.2, 0.3\}$
- **Learning rate**: $\{1 * 10^{-4}, 5 * 10^{-4}, 1 * 10^{-3}\}$
- **Exploration coefficient** $\beta$: $\{0.5, 0.1, 0.05, 0.01\}$

We run 5 seeds for each hyperparameter combination and choose the best combination based on the performance on an unreachable validation context set. For the main results we run the best hyperparameters on different seeds and evaluate on a distinct unreachable test set. A full list of parameters we used can be found in Table 3 for DQN and Table 4 for PPO.

The hyperparameter $K$ for Explore-Go that determines the maximum number of steps is chosen by a small hyperparameter search over the values: $\{25, 50, 100\}$ (for context, the episode times out after 100 steps). However, in Figure 12 we show that the performance of Explore-Go is not very sensitive to the exact value of $K$. All other hyperparameters for the Explore-Go agents are fixed to those values found for the baseline agents. The Explore-Go specific hyperparameters we use can be found in Table 5.

EXPLORE-GO, DQN, TEE AND ADDITIONAL METRICS

For the experiments comparing Explore-Go with DQN and TEE, we use the same hyperparameters as for the other DQN experiments. TEE essentially works by assigning a different $\beta_i$ to each rollout worker $i$. We assign values of $\beta_i$ according to the relation in Jiang et al. (2023): $\beta_i = \varphi \lambda^{1 + \frac{i}{N-1} \alpha}$. The term $\lambda^{1 + \frac{i}{N-1} \alpha}$ is always smaller than 1. So, we set $\varphi$ to the highest value of $\beta$ we considered in the hyperparameter search for the DQN baseline ($\varphi = 0.5$) and perform an additional hyperparameter search over the following combinations of values for $\lambda$ and $\alpha$ (see in Figure 7 how those value of $\lambda$ and $\alpha$ translate to values of $\beta_i$ for the workers):

- $\lambda$: $\{0.1, 0.5, 0.9\}$
- $\alpha$: $\{10, 25, 50\}$

For our experiments, we chose $\lambda = 0.9$ and $\alpha = 25$ based on the highest validation performance.

When comparing Explore-Go, DQN and TEE we introduce four new metrics. The first measures the fraction of state-action space that is explored. This is calculated by enumerating all possible

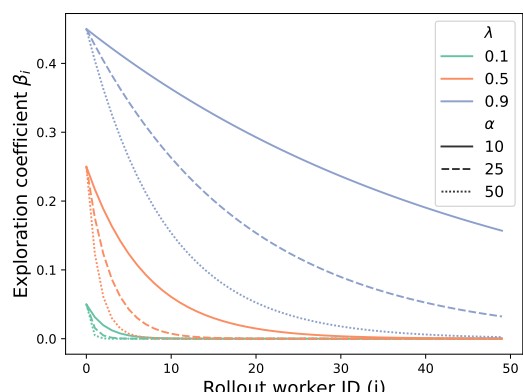

Figure 7: Exploration coefficients $\beta_i$ for 50 rollout workers for different values of TEE hyperparameters $\lambda$ and $\alpha$.

state-actions in the reachable state space and keeping track of which ones are encountered at some point during training. This measures how effective the exploration approach is (a higher fraction means the agent explored more state-actions). The second metric measures the diversity present in the replay buffer throughout training. We do so, by enumerating all possible states in the reachable space and checking which ones are present in the buffer at that time. The last metric measures the approximation error between the agent's state-value estimate $\text{argmax}_a q_\theta(s, a)$ and the true optimal value $V^*(s)$. We enumerate again over the entire reachable state space and average the approximation error. The Four Room experiments were executed on a computer with an NVIDIA RTX 3070 GPU, Intel Core i7 12700 CPU and 32 GB of memory. A single PPO or DQN run would take approximately 2 hours.

Table 2: Hyper-parameters for the architecture in the Four Rooms experiment

| CNN | |
| --- | --- |
| Kernel size | 3 |
| Stride | 1 |
| Padding | 1 |
| Padding mode | Circular |
| Channels | 64 |
| **Max Pool 2D** | |
| Kernel size | 3 |
| Stride | 2 |
| Padding | 1 |
| **Activation Function** | |
| Activation function | ReLU |

## C.3 VIZDOOM

For the ViZDoom experiment we adapt the My Way Home scenario so the goal location (location of the armour) changes between episodes (in the original My Way Home, only the agent location and orientation change between episodes). We train on a version of this My Way Home CMDP where the goal location can only be sampled from a set of 4 positions, spread-out evenly in the environment. During testing, the goal location can be sampled from 14 different positions (also spread-out roughly evenly in the environment). Since the agent cannot move the goal location, this guarantees the testing episodes are in unreachable contexts.

For our baseline APPO agent, we use the default architecture and hyperparameters from Sample Factory (Petrenko et al., 2020). For the separate, pure exploration E3B agent, trained in parallel in Explore-Go, we additionally use the architecture and hyperparameters from (Henaff et al., 2022). We

Table 3: Hyper-parameters for Four Rooms DQN

| Four Rooms DQN | |
| --- | --- |
| **Hyper-parameter** | **Value** |
| Total timesteps | 8 000 000 |
| Vectorised environments | 50 |
| Buffer size | 500 000 |
| Batch size | 256 |
| Discount factor $\gamma$ | 0.99 |
| Max. gradient norm | 1 |
| Gradient steps | 1 |
| Train frequency (steps) | 50 |
| Target update interval (steps) | 50 |
| Target soft update coefficient Q | 0.05 |
| Target soft update coefficient U | 0.005 |
| E-greedy exploration initial $\epsilon$ | 1 |
| E-greedy exploration final $\epsilon$ | 0.01 |
| E-greedy exploration fraction $\epsilon$ | 0.125 |
| Intrinsic exploration coefficient $\beta$ | 0.01 |
| **Adam** | |
| Learning rate Q | $5 \times 10^{-4}$ |
| Learning rate U | $1 \times 10^{-3}$ |

Table 4: Hyper-parameters for Four Rooms PPO

| Four Rooms PPO | |
| --- | --- |
| **Hyper-parameter** | **Value** |
| Total timesteps | 8 000 000 |
| Vectorised environments | 50 |
| Batch size | 256 |
| Discount factor $\gamma$ | 0.99 |
| Max. gradient norm | 0.5 |
| # of epochs | 5 |
| # steps collected per rollout | 12 800 |
| Entropy coeff | 0.01 |
| Value function coeff | 0.5 |
| GAE coeff $\lambda$ | 0.95 |
| Share feature extractor | True |
| Clip range | 0.2 |
| Exploration coefficient $\beta$ | 0.01 |
| **Adam** | |
| Learning rate | $5 \times 10^{-4}$ |

used a maximum pure exploration duration $K = 200$, which we chose to be slightly less than half the maximum episode duration (which times-out after 525 steps).

For the final results we train on 10 seeds (that generate the agent location, orientation and goal location from the 4 possible goal locations during training) and evaluate on the full testing distribution (which consists of all possible agent locations, orientations and goal locations from the 14 possible goal locations during testing). We calculate the mean and 95% confidence interval over 20 seeds by evaluating the agent periodically on 100 episodes in the training and testing contexts. The code can be found at <redacted for review>.

The ViZDoom experiments were executed on a computer with an NVIDIA RTX 3070 GPU, Intel Core i7 12700 CPU and 32 GB of memory. A single run would take approximately 4 hours.

Table 5: Hyper-parameters for Explore-Go in Four Rooms

| DQN | |
|---|---|
| **Hyper-parameter** | **Value** |
| Max pure exploration steps $K$ | 50 |
| **PPO** | |
| **Hyper-parameter** | **Value** |
| Max pure exploration steps $K$ | 50 |
| Pure exploration agent exploration coefficient $\beta_{\text{pure}}$ | 0.1 |

## C.4 DEEPMIND CONTROL SUITE

For the DeepMind Control Suite we adapt the environment so that at the start of each episode the initial configuration of the robot body and target location are drawn based on a given list of random seeds. This allows us to control the context space of the environment so that we can define a limited set of contexts on which the agent is allowed to train. To compute mean performance and confidence intervals we average our DMC experiments over 10 seeds for image-based experiments, and 30 seeds for the state-based Finger Turn and Reacher experiments respectively. Each agent seed trains on its own set of training contexts. For a training set of size $N$, agent $i$ gets to train on contexts generated with seeds $\{i * N, i * N + 1, ..., i * N + N - 1\}$. Testing is always done on 100 episodes from the full distribution. For the state-based experiments we train on $N = 5$ training contexts and for the image-based experiments, we train on $N = 30$ training contexts. The code can be found at `<redacted for review>`.

The standard DMC benchmark has no terminal states and instead has a fixed episode length of 1000 after which the agent times out. However, for the Finger Turn and Reacher environments, an episode length of 1000 is unnecessarily long. For these two environments, the goal is to position the robot body in such a way that some designated part is located at a target location. Once it successfully reaches this target location, the optimal policy is to do nothing. This means that in many of the Finger Turn and Reacher episodes, the agent only moves in the first 100 or so steps and then does nothing for 900 more. To simplify the training on these environments a bit we instead shorten the episode length to 500.

For the state-based experiments, we use the Explore-Go and SAC implementation adapted from Stable-Baselines3 (Raffin et al., 2021). The hyperparameters for SAC are taken from (Zhu et al., 2020). For the image-based experiments, we add Explore-Go to the RAD implementation from (Hansen & Wang, 2021) and use the hyperparameters from (Laskin et al., 2020). For all DMC experiments, we use a maximum pure exploration duration $K = 200$. We judged this to be high enough to generate diverse states in most environments.

The DMC experiments were executed on a computer with an NVIDIA RTX 3070 GPU, Intel Core i7 12700 CPU and 32 GB of memory. A single run would take approximately 13 hours.

## D PSEUDO-CODE

**Algorithm 1:** Generic CollectRollouts + Explore-Go

---

**Input:** number of steps to collect $N$, pure exploration policy $\pi_{PE}$, max number of pure
  exploration steps $K$

$k \leftarrow Uniform(0, K)$;
$\mathcal{D}_{rollout} \leftarrow \{\}$;
$num\_steps\_collected \leftarrow 0$;
**while** $num\_steps\_collected < N$ **do**
  **if** $episode\_step < k$ **then**
  | Sample transition $t$ using $\pi_{PE}$;
  **else**
    Sample transition $t$ with base RL algorithm;
    Add $t$ to $\mathcal{D}_{rollout}$;
    $num\_steps\_collected$ += 1;
  **end if**
  $episode\_step$ += 1;
  **if** $end\ of\ episode$ **then**
    $k \leftarrow Uniform(0, K)$;
    $episode\_step \leftarrow 0$;
    Reset environment;
  **end if**
**end**
Return $\mathcal{D}_{rollout}$;

---

Figure 8: An example of pseudo-code for Explore-Go for a given pure exploration policy $\pi_{PE}$ when combined with a generic rollout collection function found in some form in most RL algorithms.

## E  ADDITIONAL EXPERIMENTS

### E.1  EXPLORE-GO ON CONTINUOUS CONTROL

To further demonstrate the scalability and generality of our approach we evaluate Explore-Go on some of the continuous control environments from the DeepMind Control Suite (Tassa et al., 2018). In the DMC environments, at the start of every episode, the initial configuration of the robot body (and in some environments, target location) is randomly generated based on some initial seed. Typically, the DMC benchmark is not used for the ZSPT setting and training is done on the full distribution of contexts (initial configurations). To turn the DMC benchmark into an instance of the ZSPT problem, we define a limited set of seeds (and therefore initial configurations) on which the agents are allowed to train. We then test on the full distribution. Note that only the following environments would have unreachable testing contexts (due to varying target locations that the agent cannot modify during an episode): Reacher, Finger Turn, Manipulator, Stacker, Fish and Swimmer. For the other environments, all contexts are reachable from one another. See Supplement C.4 for more details on these experiments.

The DMC environments can be explored sufficiently with $\epsilon$-greedy exploration, which means for Explore-Go we used a uniformly random action selection policy (equivalent to setting $\epsilon = 1$) during the pure exploration phase. We run two experiments on the Finger Turn Easy and Reacher Easy environments: one with a short vector-based state, and one with images as observations. Experiments with states as observations using SAC as a baseline show only a slight advantage of Explore-Go over baseline SAC, with questionable significance (left in Figure 10). However, the same experiment on corresponding image observations (right in Figure 10), shows a significantly improved generalisation performance by Explore-Go, at least for Finger Turn Easy. Here we use as a baseline RAD, which consists of SAC with random image cropping to increase generalisation.

**Algorithm 2:** PPO + Explore-Go

**Input:** PPO agent $PPO$, pure exploration agent $PE$, max number of pure exploration steps $K$
$k \leftarrow Uniform(0, K)$;
$i \leftarrow 0$                $\triangleright$ Counts steps within an episode;
**for** $iteration = 0, 1, 2, ...$ **do**
    $\mathcal{D}_{PPO} \leftarrow \{\}$;
    $\mathcal{D}_{PE} \leftarrow \{\}$;
    **for** $step = 0, 1, 2, ..., T$ **do**
        **if** $i < k$ **then**
            Sample transition $t$ by running $PE$;
            Add $t$ to $\mathcal{D}_{PE}$;
        **else**
            Sample transition $t$ by running $PPO$;
            Add $t$ to $\mathcal{D}_{PPO}$;
        **end if**
        $i \leftarrow i + 1$;
        **if** *end of episode* **then**
            $k \leftarrow Uniform(0, K)$;
            $i \leftarrow 0$;
            Reset environment;
        **end if**
    **end**
    Update $PPO$ with trajectories $\mathcal{D}_{PPO}$;
    Update $PE$ with trajectories $\mathcal{D}_{PE}$;
**end**

Figure 9: An example of pseudo-code for Explore-Go combined with an on-policy method PPO when the pure exploration agent $PE$ has to be trained on the pure exploration experience in parallel to the main agent.

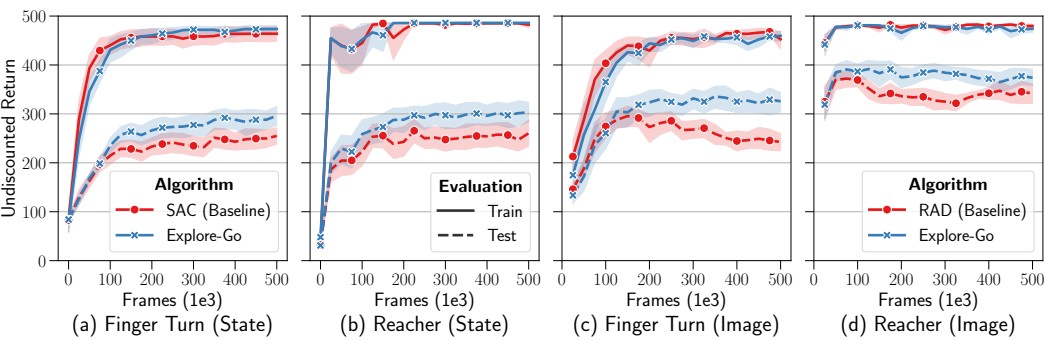

(a) Finger Turn (State)      (b) Reacher (State)      (c) Finger Turn (Image)      (d) Reacher (Image)

Figure 10: DMC performance of Explore-Go with (a & b) SAC on the underlying state and (c & d) RAD on image observations. Shown are the mean and 95% confidence intervals over 30 seeds for the state-based experiments and 10 seeds for the ones based on images.

### E.2 Adding pure exploration experience to the buffer

In Figure 11 we show an ablation of Explore-Go where we also add all the pure exploration experience to the replay buffer (Explore-Go with PE, green). It shows that adding this experience to the buffer increases the value error and decreases the generalisation performance.

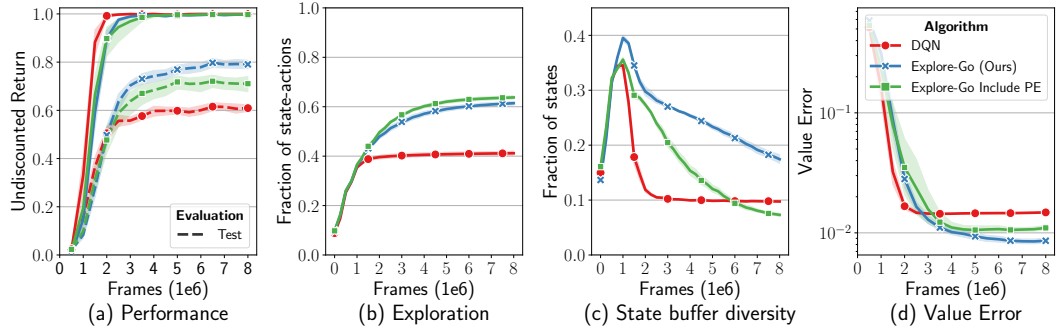

(a) Performance  (b) Exploration  (c) State buffer diversity  (d) Value Error

Figure 11: Comparing DQN, DQN+Explore-Go, and DQN+Explore-Go with pure exploration experience added to the buffer (Explore-Go with PE) in Four Rooms: (a) train and test performance, (b) fraction of reachable state-action space explored, (c) fraction of reachable state space currently in the buffer and (d) true value error averaged of the entire reachable state space. Shown are the mean and 95% confidence intervals over 20 seeds.

### E.3 SENSITIVITY STUDY $K$

In Figure 12 we show a sensitivity study of the Explore-Go hyperparameter $K$ that determines the maximum number of pure exploration steps per episode. It shows that the performance of Explore-Go is not very sensitive to the exact value of $K$, but does show a trend where performance deteriorates as $K$ gets too small or too high. This can be explained by the fact that if $K$ is too small, Explore-Go will not perform enough pure exploration steps at the start of every episode, and won't induce very diverse new starting states. On the other hand, if $K$ is too large, Explore-Go will spend most of the episode purely exploring, which means it will train on relatively little data (significantly hurting sample efficiency).

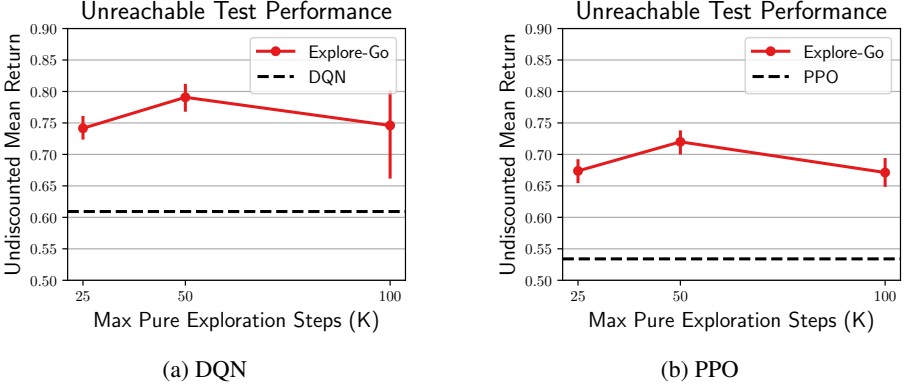

(a) DQN  (b) PPO

Figure 12: Unreachable generalisation performance after training for different values of the Explore-Go hyperparameter $K$ that determines the maximum number of pure exploration steps per episode for (a) DQN and (b) PPO. The dashed line is the final test performance of the baseline. Shown are mean and 95% confidence intervals over 20 seeds.

