# OpenReview forum: "Exploration Implies Data Augmentation: Generalisation in Contextual MDPs"
_ICLR.cc/2026/Conference — Submitted to ICLR 2026_

### Official Review · Reviewer_SCyZ · 2025-10-28

**Soundness:** 3
**Presentation:** 3
**Contribution:** 2
**Rating:** 4
**Confidence:** 4

**Summary:**

This paper studies the problem of zero-shot generalization in contextual Markov decision processes (CMDPs). In particular, the problem is investigated through the effects of exploration on aiding the generalization of RL agents. The authors demonstrate that training on more states that are reachable (i.e., those with non-zero probability of being encountered during training) can improve generalization to unreachable states during evaluation. Then, a method named Explore-Go is introduced, which consists of increasing the diversity of the starting state distribution by, at the start of every training episode, following a pure exploratory policy for a random number of steps, and then actually starting the episode from the state the agent ended in. Experiments were conducted in the FourRoom, ViZDoom, and DMC, and compared the performance and value error of the baseline method with that of the Temporally Equalised
Exploration (TEE) in the FourRoom domain.

**Strengths:**

- The paper is well-structured and well-written (easy to follow).
- The proposed method is simple and easily applicable, making it possible to combine with different methods that tackle orthogonal problems.

**Weaknesses:**

- Although the proposed technique improves test performance, it has been shown to sometimes decrease performance in the context of the training distribution.
- The setting only considers contextual MDPs that do not have context-conditioned transitions and rewards. Although the authors state that this is the case in many benchmarks, the assumption that the context is observable in $s$ can be very restricting.
- Although Section 3 discusses the problem intuitively, the paper could be strengthened with a more formal/mathematical analysis of the proposed method. For instance, what is the impact of the number of random steps $K$ on the generalization capabilities of the agent? How much does the method reduce the impact of inaccurate targets?
- Most of the experimental analysis focused on the simpler FourRoom domain. Moreover, the results in the DMC and ViZDoom benchmarks did not include the competing state-of-the-art baseline TEE.
- The gains in test performance are not that significant in some tasks (e.g., DMC, see Fig. 10).

**Questions:**

Below, I have a few questions/additional comments:

- Why can TEE not be used with on-policy approaches?

- “Any CMDP where the context is observable in s′ is in this subset, but multiple contexts can also share states s′.” This is not very clear to me. Do the authors mean that you are considering states in the form $s’=(s,c)$? In this case, T(s’, c’, a) = T(s’, a), but $c’ \neq c$? I suggest clarifying this, perhaps with a clear example.

- “Line 187: in which case they will never share the same underlying state”. If contexts never share the same underlying state, why consider a CMDP formulation? It seems to me that each context is actually a different MDP.

Minor:

- Line 92: The common convention is to use parentheses instead of brackets for defining tuples.

- Line 291: “To improve generalization by both training on additional states and providing accurate targets for these states, we propose them as the starting states of additional training contexts.” What is “them” in this sentence?

---

> ### Author Response · Authors · 2025-11-20
>
> We appreciate the reviewer's feedback and questions, and address each major point below:
> - **Decreased Performance in the Training Contexts:** Explore-Go can indeed sometimes result in worse performance on the training contexts. However, in the zero-shot policy transfer setting the most important metric is the performance in the testing contexts. By achieving lower training performance, but _higher_ testing performance, Explore-Go seems to overfit _less_ to the training contexts, which we consider a strength rather than a weakness.
> - **Context-Conditioned Transitions and Rewards:** We clarify that our definition of a CMDP subclass does encompass context-conditioned transitions and rewards, provided the observation the agent sees $\phi(s)$ is Markovian. We clarify this point more in our general comment addressed to all reviewers.
> - **Limited Experimental Analysis and Missing TEE Baseline:** We acknowledge the focus on Four Rooms for the detailed analysis. This choice was dictated by the large computational expense of performing exhaustive comparisons, including TEE, across the slower DMC and ViZDoom environments. Importantly, TEE is fundamentally an off-policy method and cannot be used as a competing baseline for our ViZDoom results, which employ the on-policy APPO algorithm, which further highlights the generality of Explore-Go over TEE.
> - **Why can TEE not be used with on-policy approaches?** TEE's core principle, maintaining extended, non-decaying exploration throughout training, directly conflicts with the principles of on-policy optimisation. On-policy methods must decay intrinsic rewards to zero to ensure the final policy is optimised solely for the original extrinsic objective. Furthermore, TEE relies on training from trajectories collected by multiple, concurrently running policies ($\pi$ with different $\beta$'s), which is structurally incompatible with the on-policy training update mechanism, which requires training data collected by a single policy.
> - **"Any CMDP where the context is observable in s' is in this subset, but multiple contexts can also share states s'."** The reviewer is correct that this point requires clearer explanation. We attempt to clarify this in our general comment addressed to all reviewers.
> - **If the contexts never share the same underlying state, why consider the CMDP formulation?** We agree that when contexts never share the same underlying state, the CMDP effectively becomes a set of distinct MDPs. To some extent, this is true for any CMDP. However, the example for this setting that we provide in the paper, where each context is a different maze, is often viewed through the CMDP formulation. So, we believe using the established CMDP framework is useful for relating our analysis to the broader zero-shot generalisation literature.
> - **Line 92: Typographical Error:** We acknowledge the error regarding tuple notation (using brackets instead of parentheses) and will correct it in the final paper.
> - **Line 291: Ambiguity in "them":** The word "them" refers to the "additional states" encountered during exploration. We will rewrite the sentence to clearly indicate that we propose using these additional states as new starting points for training contexts.
>
> We appreciate all the feedback from the reviewer and commit to using this feedback and the subsequent discussion to improve the clarity of the final version of the paper.

---

> > ### Comment · Reviewer_SCyZ · 2025-11-24
> >
> > I thank the authors for their response. I would like to maintain my score since I still have shared concerns with the other reviewers (e.g.,  Four Rooms as the only environment where Explore-Go is contrasted with exploration-heavy baselines).

---

### Official Review · Reviewer_PzCb · 2025-10-29

**Soundness:** 2
**Presentation:** 4
**Contribution:** 2
**Rating:** 4
**Confidence:** 3

**Summary:**

The authors introduce, identify, and address an issue with misled value targets in reinforcement learning policies when exploring. The proposed method holds out on training the policy in the first (uniformly sampled) $k$ steps of an episode. Meanwhile, a separate exploration policy guides the agent. Transitions for training only include data after the exploration phase ends at $k$.

**Strengths:**

The introduced method appears* simple, effectively introducing an alternative online policy which is dedicated to exploring the environment, and adding additional data to the experience replay.

*the exploration policy does not appear to be defined in detail. See questions.

**Weaknesses:**

My main questions seek a bit more clarity on the core assumptions and experimental design. I'd appreciate a clearer justification for the exploration problem as framed, and a bit more high-level context for the foundational concepts built upon, like Jiang et al. and $\pi_{PE}$. My primary concern is that the main experiment's training intervention (more start states) seems to directly correlate with the test evaluation (new start states), so I would appreciate clarity on what other axes of generalization are being measured. Please see my questions.

**Questions:**

“Recently Jiang et al. (2023) demonstrated that continuous exploration throughout training can improve
generalisation to test contexts. One of their core algorithmic contributions, temporally equalised
exploration (TEE), assigns different exploration coefficients β
… To separate the effects of continuous and better exploration, this paper uses a TEE variant where parallel workers follow count-based intrinsic reward with different β.”
It seems this work is referenced several times (lines 220, 263) throughout the work and appears to be a central foundation on which the submitted work relies. If this is the case, it is certainly deserving of more high-level explanation and summary than the technical brief that is given.
For example, elaboration for “intrinsic reward with different \beta” would be helpful (or a brief aside indicating that it is not crucial to understand).

What exactly is $\pi_{PE}$? The only explanation I can find for it is in lines 302-304, which gives a _very_ high-level definition. However, it is used in the pseudocode for Explore-Go in the appendix. Is it uniformly random? Is it entropy or information maximizing? Exploration is incredibly nuanced and its exact definition in $\pi_{PE}$ needs to be made clear in order to understand the implications of the experiments.

In the experiments, (a) the training environment is a grid-world-like environment in which the objective is to move from starting location to end location. (b) In the test environment, only environments with different starting and ending goals from the training environment are used. (c) Training with Explore-Go effectively only changes the distribution of starting states. Then, is it not obvious that a method which trains agents on more starting locations will certainly improve performance? Since, augmenting starting locations is directly correlated with the generalization evaluation. I feel that I must be missing something – I would appreciate clarification on what other axes of generalization are being evaluated in this experiment. Without this, I am afraid I do not see how the results move beyond the straightforward general machine learning hypotheses of “training on data closer to the test data will improve test scores.” Is there perhaps another experiment environment that might de-correlate the results with the coverage of starting states? For instance, a locomotion task where the contexts are within some friction range and the unreachable context lies outside this range?

In figure 4d, what does $\beta=0.5$ eventually come down to, provided enough frames?

“To get an accurate estimate of Qπ (s′, a′), we need to observe the consequences of choosing a′ in s′, but exploration often picks actions the policy would not. Therefore, bootstrapping from state-actions that are not trained on makes the targets of many exploratory transitions inaccurate.” It is certainly possible to model Q as a distribution such that the policy is picking actions from a distribution created by normalizing Q while it is still uncertain about its actions. In this case, sampling the policy and estimating Q is exactly the same. I believe what the authors mean to argue is perhaps a randomly exploring policy where its actions are completely unrelated to the task. If this is apparent from following Jiang et al., I would again encourage authors to dedicate more material to explaining (or reproducing and reframing) that reference before.

---

> ### Author Response · Authors · 2025-11-20
>
> We appreciate the reviewer's feedback and questions, and address each major point below:
> - **Elaboration on TEE (Jiang et al., 2023):** We recognise the discussion of TEE is perhaps too brief for its importance to the paper. The core idea of TEE is to ensure continuous exploration throughout training by assigning a different, fixed value of the exploration constant $\beta$ to each parallel rollout worker. While most methods finely tune $\beta$ so that exploration decays at precisely the right time towards the end of training, TEE assigns some workers a much higher $\beta$, preventing exploration from stopping entirely. We will improve the clarity of this approach in the main text.
> - **Clarification of Pure Exploration Policy (**$\pi_{PE}$**):** The policy $\pi_{PE}$ is a pure exploration policy that ignores extrinsic rewards $r$ and optimises only the intrinsic reward $\eta$. The exact formulation of $\eta$ (e.g., novelty, uncertainty, entropy) determines the specific exploration behaviour. Since Explore-Go focuses on _when_ and _how_ exploration is used rather than _which_ exploration technique is best, we consider it largely orthogonal to the exact choice of $\eta$ (even though it is true that different choices can result in different generalisation performance, that is not the focus of this paper). We will expand the definition of $\pi_{PE}$ in the main text and add discussion explaining that the specific exploration technique is not the focus of this paper.
> - **Generalisation Axis and Correlation with Starting States:** This is a crucial observation, and we believe the reviewer might be missing the distinction we make in Section 3.1 and Appendix B between _reachable_ and _unreachable_ generalisation.
>     - **Reachable Generalisation** (e.g., same goal, different starting position) is trivially improved by training on more starting states.
>     - **Unreachable Generalisation** (e.g., new goal location, which fundamentally changes the optimal policy) does _not_ trivially improve by merely training on more starting states from the original contexts.
>     - All our experiments evaluate generalisation to unreachable contexts (e.g., new goal or doorway configurations). We are the first to explicitly argue that training on more reachable starting states (same goal, different starting position) can _also_ improve generalisation to these _unreachable_ contexts (new goal location), which is the core novel hypothesis of our paper.
> - **DQN** $\beta = 0.5$ **in Figure 4d over more frames:** We will perform an extended run of the DQN baseline with $\beta = 0.5$ and include the result in the final paper. However, we argue that even if it were to achieve comparable performance to Explore-Go, Figure 4 already clearly demonstrates that this approach is significantly less sample efficient, highlighting a key trade-off that Explore-Go is designed to mitigate.
> - **Bootstrapping and Policy Distribution:** We acknowledge the nuance regarding modelling $Q$ as a distribution. Our argument applies to any exploration approach where the $\pi_{PE}$ policy optimises an alternative objective (intrinsic reward) that is in some sense orthogonal to the extrinsic task reward. When calculating $Q^\pi$ targets using off-policy transitions generated by $\pi_{PE}$, the targets for the primary, task-solving $\pi$ become unreliable due to the policy mismatch, even if the policy itself is stochastic. This can of course be solved by collecting more data, which is why we argue this problem mainly exists for exploratory states, which are by definition rarely visited.
>
> We appreciate all the feedback from the reviewer and commit to using this feedback and the subsequent discussion to improve the clarity of the final version of the paper.

---

### Official Review · Reviewer_SAMA · 2025-11-01

**Soundness:** 2
**Presentation:** 3
**Contribution:** 2
**Rating:** 4
**Confidence:** 3

**Summary:**

This paper proposes Explore-Go, a simple training procedure that improves generalization in contextual Markov decision processes (CMDPs) by expanding the coverage of initial states. Concretely, each episode begins with a short exploratory prefix to reach a diverse set of states; learning then proceeds from these reached states as if they were new “starts.” The method is evaluated on Four Rooms, DeepMind Control (DMC), and VizDoom, where it shows consistent gains over standard baselines.

**Strengths:**

1. Rich quantitative analysis. The experiments track multiple metrics, yielding conclusions that feel natural rather than cherry-picked.
2. Simplicity and practicality. The idea is straightforward and easy to implement in common RL codebases.
3. Thorough ablations. The paper includes many ablation studies that clarify which components matter.
4. Strong empirical results. Within the tested settings, the method reaches or matches state-of-the-art performance.

**Weaknesses:**

1. Over-narrow CMDP setting (under-signaled).
   The paper effectively restricts to a CMDP subclass where context only alters the initial-state distribution—a perfect match to Explore-Go’s design. This specialization is neither reflected in the title nor called out clearly in Section 2’s definition, which risks misinterpretation. As a result, the method can feel tailored to this specific subclass; its general significance is under-argued. The paper should explain why this CMDP subclass is important in practice.

2. Comparisons are limited beyond Four Rooms.
   On DMC and VizDoom, the method is compared mainly against SAC, RAD, and APPO, which do not emphasize exploration. This leaves Four Rooms as the only environment where Explore-Go is contrasted with exploration-heavy baselines, making it harder to claim superiority in broader exploration settings. Given the paper’s already narrow problem scope, and that baselines like TEE are applicable more widely, the comparative value of these experiments is reduced.

3. Sample-efficiency cost is acknowledged but unquantified.
   The method discards exploratory prefixes for training targets, which likely hurts sample efficiency. The paper notes this qualitatively but does not offer a direct, controlled measurement (e.g., equal wall-clock, equal updates, equal environment steps) to show the true cost-benefit trade-off.

4. Key insights remain largely intuitive.
   Several claims—such as how exploration mitigates spurious correlations—are supported mainly by intuition and a toy example in Section 3. That example, however, relies on extremely low coverage relative to state dimensionality, which is highly constructed and may not reflect realistic regimes. Stronger validation in more complex settings is needed to substantiate these insights.

**Questions:**

1. Why modify the default reward in Four Rooms for the experiments? What behavior or evaluation property required this change?
2. A substantial portion of the paper motivates exploration and its trade-offs, which is relatively familiar. The crucial unresolved point is why the chosen CMDP subclass (context alters only the start distribution) is broadly important. Could you add more environments to demonstrate that many practical tasks fit this subclass, thereby strengthening the case?
3. Some legend placements are confusing—for example, in Figure 4, “Evaluation” appears in (a) while “Algorithm” appears in (d). Could you standardize or clarify legend placement?
4. In Figure 4, the DQN baseline appears undertrained. With sufficient training time or a tuned schedule, does DQN approach Explore-Go’s performance? If not, what concrete barrier remains?

---

> ### Author Response · Authors · 2025-11-20
>
> We appreciate the reviewer's feedback and questions, and address each major point below:
> - **Comparisons are limited beyond Four Rooms:** We agree that the TEE baseline could have also been combined with SAC and RAD in the DMC environments. However, one of the main contributions of our approach Explore-Go is that it can also be combined with on-policy approaches, which is not true for approaches like TEE. As such, we believe we already demonstrate compelling arguments in favour of Explore-Go, by demonstrating it outperforms TEE in Four Rooms _and_ works with various different algorithms, including on-policy ones.
> - **Sample-efficiency cost is acknowledged but unquantified:** We would like to clarify that the cost of exploration _is_ in fact quantified and accounted for in all our figures. The x-axis, labelled "Frames," includes _all_ steps taken in the environment, including the transitions collected during the pure exploration phase. This ensures all comparisons are made at equal environment steps taken, which we believe is the fairest metric.
> - **Key insights remain largely intuitive:** We agree that the foundation for our hypothesis is built on the illustrative CMDP in Figure 2. However, we would argue that the quantitative results in the Four Rooms environment (Figure 4), which show Explore-Go achieving superior performance with better value accuracy than continuous exploration methods (TEE, DQN $\beta=0.5$), provide some of the necessary empirical support for our core motivation. Furthermore, we believe that low coverage relative to state dimensionality is actually a problem that tends to become worse the more realistic and complex the environment gets.
> - **Why modify the default reward in Four Rooms for the experiments?** We made a minor change to align the reward function with the common convention of 1 for successfully reaching the goal state and 0 otherwise. This ensures the undiscounted return directly equals the success rate, simplifying the interpretation of the evaluation metric without producing any noticeable change in the agent's behaviour.
> - **The crucial unresolved point is why the chosen CMDP subclass... is broadly important:** We believe the chosen CMDP setting (context only alters start distribution) is quite general, as it encompasses any CMDP where the observation the agent sees $\phi(s)$ is Markovian. We clarify this point more in our general comment addressed to all reviewers.
> - **Some legend placements are confusing** We apologise for the confusing placement. We commit to improving the standardisation and placement of all legends in the final version of the paper to ensure clarity while minimising obstruction of the figure data.
> - **In Figure 4, the DQN baseline appears undertrained** In Appendix C.2 we clarify that the DQN baseline was tuned for the given training budget (8M steps) and Figure 4 depicts the DQN hyperparameters that achieved maximal validation performance. While DQN might achieve similar final performance to Explore-Go with a significantly larger budget in this small environment, Figure 4 already illustrates that increasing the exploration coefficient ($\beta=0.5$) for the baseline makes it substantially less sample efficient than Explore-Go. This suggests that achieving high performance without the Explore-Go mechanism requires an inefficient use of resources.
>
> We appreciate all the feedback from the reviewer and commit to using this feedback and the subsequent discussion to improve the clarity of the final version of the paper.

---

### Official Review · Reviewer_sgXQ · 2025-11-01

**Soundness:** 2
**Presentation:** 1
**Contribution:** 3
**Rating:** 4
**Confidence:** 4

**Summary:**

The paper proposes to use exploration as a starting phase to increase generalization in contextual MDPs. The authors motivate this idea as a form of augmentation - training on more initial states artificially increases the number of contexts, which can potentially break spurious correlations that usually occur when training on a small number of contexts. Moreover, the paper argues that a naive application of exploration can reduce the accuracy of the learned value function, thereby hindering generalization. To address this, the proposed algorithm Explore-Go demonstrates improved generalization performance across several environments.

**Strengths:**

* The idea of using exploration for data augmentation to generate more contexts (tasks) in CMDP is new and well-motivated. I particularly like the illustrative CMDP task in Figure 2, which clearly explains the motivation for the proposed method.

* The paper provides a thorough discussion on related work on exploration for generalization and their connection to this work.

* The experiments section contains 3 different domains, providing diverse evaluation scenarios.

**Weaknesses:**

* The writing and presentation could be improved. For example, Figure 2's illustration of how exploration improves generalization could be moved to the introduction to better motivate the method and engage readers earlier.

* The reasons why naive exploration reduces the accuracy of the learned value function are not clear to me.

**Questions:**

* The problem of wrong targets (inaccurate value function): I completely agree that incorporating state space exploration in the training process as an adaptive intrinsic reward to the extrinsic reward (as done in the SOTA TEE), leads to training on wrong targets- it depends on the weight coefficient between the intrinsic and extrinsic rewards if the target is to continue explor or to exploit (reach to the task’s goal). This mix of two contradicting rewards can lead the agent to learn the wrong target (continue to explore when it should exploit).  However, why this leads to an inaccurate value function is not clear to me. Could you clarify this point?

* When the value error in Figures 1 and 4 is calculated, is it the error between the extrinsic reward (what should be the true reward) and the intrinsic + extrinsic reward (for TEE and DQN with β = 0.5)?

* In Figures 1 and 4 - Why is the value error of vanilla DQN higher than Explore-Go?

* Do you use memory in the initial exploration phase of Explore-Go?

* How would this method work for non-navigation tasks when the training environments have unrecoverable states? I assume the training performance decreases, but how will it affect the test and generalization performance?

* For future work: an agent trained for pure exploration is known to exhibit less task-specific behavior that generalizes to unseen tasks (such as following the “right-hand rule” for solving mazes). Did you try combining pure exploration with Explore-Go at test time or as an initialization step? I wonder if pure exploration could improve generalization beyond simply providing data augmentation that breaks spurious correlations - perhaps by learning a more general behavior.

More writing suggestions: consider reducing the number of footnotes and integrating them into the main text to make the reading more fluent.

---

> ### Author Response · Authors · 2025-11-20
>
> We appreciate the reviewer's feedback and questions, and address each major point below:
> - **The Problem of Wrong Targets (Inaccurate Value Function):** We confirm that combining intrinsic and extrinsic rewards results in learning targets that are inaccurate with respect to the original task. However, our argument for an inaccurate value function differs from this:
>     - In Q-learning, the value $Q^\pi(s,a)$ for a state $s$ and action $a$ is trained on a bootstrapped estimate of the value $\mathbb{E}^{a' \sim \pi} [Q^\pi(s',a')]$ in the next state $s'$. However, exploration tends to choose actions that the policy $\pi$ would not/rarely choose, since its objective is to uncover _novel_ states. But, if the state $s$ and $s'$ were only encountered during exploration, the agent will likely not have evidence for what happens in state $s'$ if it would follow policy $\pi$. Therefore, for exploratory states, that are infrequently visited, or visited only by an exploratory policy, the bootstrapped targets  $\mathbb{E}_{a' \sim \pi} [Q^\pi(s',a')]$ will likely be inaccurate. We believe the results for DQN with increased exploration (DQN $\beta = 0.5$) in Figure 4 are consistent with this argument, as it explores the most (4b) and has the highest buffer diversity (4c), both indicating it has the most 'exploratory states', yet has significantly higher value error compared to all the other approaches. One approach to overcome this issue, is that the agent goes to a novel exploratory state, and once its there, samples from the policy $\pi$. This is effectively what Explore-Go aims to do.
> - **Value Error Calculation in Figures 1 and 4:** The value error in Figures 1 and 4 is calculated with respect to the task-specific extrinsic value function $Q^\pi$ only, ensuring a direct comparison of the agent's performance on the original task objective.
> - **Value Error of Vanilla DQN vs. Explore-Go (Figures 1 and 4):** The value error in Figure 1 and 4 is defined as the difference between the learned (extrinsic) Q-value and the true one, averaged over the _entire_ reachable state space (so including states the agent has never encountered). Vanilla DQN's error is higher because, although it has accurate targets, it trains on a smaller fraction of the reachable state space than Explore-Go.
> - **Use of Memory in Initial Exploration Phase:** Memory (Recurrent Neural Networks) is not used for the initial pure exploration phase in our fully observable MDP experiments (Four Rooms, DMC). Memory _is_ used for both the pure exploration policy and the main agent in the partially observable ViZDoom environment.
> - **Application to Non-Navigation Tasks with Unrecoverable States:** Although our paper is focused on _how_ exploration should be used, rather than _which_ exploration technique is best, it is true that the performance of Explore-Go depends to some extent on the chosen pure exploration policy ($\pi_{PE}$). In environments with many unrecoverable, low reward states, $\pi_{PE}$ would need to be augmented or biased with an objective to avoid entering such irreversible states too often. However, this is a general necessity for any effective exploration strategy in these types of environments, not a limitation specific to Explore-Go.
> - **Pure Exploration at Test Time:** Leveraging the pure exploration policy at test time (as an initialisation step or in conjunction with the learned policy) is a promising and logical extension for future work, as Explore-Go already requires training such a pure exploration policy. We have currently not yet investigated this direction.
>
> We appreciate all the feedback from the reviewer and commit to using this feedback and the subsequent discussion to improve the clarity of the final version of the paper.

---

### Author Response · Authors · 2025-11-20
**General Comments**

Several reviewers had questions about the generality of our CMDP framework. We agree that the explanation in Section 3 could be improved, and would like to clarify our specific CMDP setting in more detail here, and argue it is not as strong a restriction as it appears at first.

In the CMDP framework, the state $s = (s',c) \in S$ can in principle be decomposed into an underlying state $s' \in S'$ and a context $c \in C$. In general, the agent does not observe $s$ directly, but rather the output of some observation function $\phi(s)$. For our analysis in Section 3, we assume a CDMP $\mathcal{M}$ where the observations $\phi(s)$ satisfy the Markov property. Note that this does not mean the agent directly observes the exact context $c$. Instead, the observation $\phi(s)$ has to directly or indirectly encode only those parts of the context that affect the rewards or dynamics.

An example of this is the Four Rooms environment (see Figure 6 in Appendix C.2). In this MDP, the context is defined by the agent's starting location, the goal location, and the location of the doorways connecting the four rooms. However, the agent only observes an image encoding similar to the one in Figure 6. This $\phi(s)$ implicitly encodes in its pixel information the parts of the context that affect dynamics (location of the doorways) and rewards (goal location), but does not encode the agent's original starting location, which is not necessary for the Markov property.

If the observations $\phi(s)$ are Markov, this means any contextual information that affects transitions or rewards has to be encoded in $\phi(s)$. Since contexts are determined before an episode starts, and do not change throughout that episode, it also means this information has to already be encoded in some way in the starting observation $\phi(s_0)$. Therefore, any CMDP $\mathcal{M}$ where the observations $\phi(s)$ are Markov, can be equivalently defined as a CMDP $\hat{\mathcal{M}}$ where the context $\hat{c} \in \hat{C}$ only determines the agent's starting state $\hat{s}' \in \hat{S}'$, where $\hat{s}' = \phi(s)$ for a state $s \in S$ in the original CMDP $\mathcal{M}$. However, as highlighted above, this _does_ also allow for context-dependend transitions and rewards. We will adapt Section 3 to follow the alternative definition of our setting we described here, as we hope this more intuitively communicates the generality and importance of our setting.

---

### Meta-Review · Area_Chair_GUPG · 2025-12-30

**Summary:**

This paper introduces Explore-Go, a method that enhances zero-shot generalization in contextual MDPs by utilizing an initial exploration phase to generate diverse starting states. Its primary strength lies in its simplicity and practical effectiveness, which demonstrate consistent empirical gains across several benchmarks and various reinforcement learning algorithms. However, reviewers raised significant concerns that comparisons with the state-of-the-art TEE baseline were limited to the Four Rooms environment, leaving its relative advantage in more complex domains unproven. Furthermore, the narrow focus on a CMDP subclass where context only modifies initial state distributions leads to doubts regarding the method's broader applicability and whether the performance gains are a trivial consequence of training on a distribution that directly matches the evaluation criteria.

**Reviewer Concerns:**

The authors provided a robust rebuttal that effectively addressed several core technical misunderstandings. Regarding the first concern, they correctly emphasized that while TEE is a strong baseline, it is fundamentally restricted to off-policy algorithms, whereas Explore-Go offers broader applicability across both on- and off-policy methods. For the second concern about the narrow CMDP subclass, the authors presented a persuasive explanation and mathematical theory, arguing that their framework encompasses any CMDP where observations satisfy the Markov property. Despite these insightful clarifications, the concern regarding the limited empirical comparison with TEE remains partially outstanding, as reviewers felt that more extensive experiments in diverse and complex environments were still necessary to fully validate the claimed superiority.

**Reviewer Scores:**

The high quality of the rebuttal provided a strong basis for potential score increases. However, Reviewer SCyZ explicitly stated an intention to maintain their score, mentioning the lack of contrast with the baselines beyond the Four Rooms environment. Even if one or two other reviewers were to raise their ratings, it is unclear whether such a shift would provide a sufficiently strong signal for acceptance. Given the current relatively low scores, I recommend Reject.

---

### Decision · Program_Chairs · 2026-01-26

Reject